# Better sampling in explanation methods can prevent dieselgate-like deception

## Abstract

Machine learning models are used in many sensitive areas where, besides predictive accuracy, their comprehensibility is also important. Interpretability of prediction models is necessary to determine their biases and causes of errors and is a prerequisite for users' confidence. For complex state-of-the-art black-box models, post-hoc model-independent explanation techniques are an established solution. Popular and effective techniques, such as IME, LIME, and SHAP, use perturbation of instance features to explain individual predictions. Recently, Slack et al. (2020) put their robustness into question by showing that their outcomes can be manipulated due to poor perturbation sampling employed. This weakness would allow dieselgate type cheating of owners of sensitive models who could deceive inspection and hide potentially unethical or illegal biases existing in their predictive models. This could undermine public trust in machine learning models and give rise to legal restrictions on their use.

We show that better sampling in these explanation methods prevents malicious manipulations. The proposed sampling uses data generators that learn the training set distribution and generate new perturbation instances much more similar to the training set. We show that the improved sampling increases the LIME and SHAP's robustness, while the previously untested method IME is already the most robust of all.

## 1 Introduction

Machine learning models are used in many areas where besides predictive performance, their comprehensibility is also important, e.g., in healthcare, legal domain, banking, insurance, consultancy, etc. Users in those areas often do not trust a machine learning model if they do not understand why it made a given decision. Some models, such as decision trees, linear regression, and naïve Bayes, are intrinsically easier to understand due to the simple representation used. However, complex models, mostly used in practice due to better accuracy, are incomprehensible and behave like black boxes, e.g., neural networks, support vector machines, random forests, and boosting. For these models, the area of explainable artificial intelligence (XAI) has developed post-hoc explanation methods that are model-independent and determine the importance of each feature for the predicted outcome. Frequently used methods of this type are IME (Štrumbelj & Kononenko, 2013), LIME (Ribeiro et al., 2016), and SHAP (Lundberg & Lee, 2017).

To determine the features' importance, these methods use perturbation sampling. Slack et al. (2020) recently noticed that the data distribution obtained in this way is significantly different from the original distribution of the training data as we illustrate in Figure 1a. They showed that this can be a serious weakness of these methods. The possibility to manipulate the post-hoc explanation methods is a critical problem for the ML community, as the reliability and robustness of explanation methods are essential for their use and public acceptance. These methods are used to interpret otherwise black-box models, help in debugging models, and reveal models' biases, thereby establishing trust in their behavior. Non-robust explanation methods that can be manipulated can lead to catastrophic consequences, as explanations do not detect racist, sexist, or otherwise biased models if the model owner wants to hide these biases. This would enable dieselgate-like cheating where owners of sensitive prediction models could hide the socially, morally, or legally unacceptable biases present in their models. As the schema of the attack on explanation methods on Figure 1b shows,

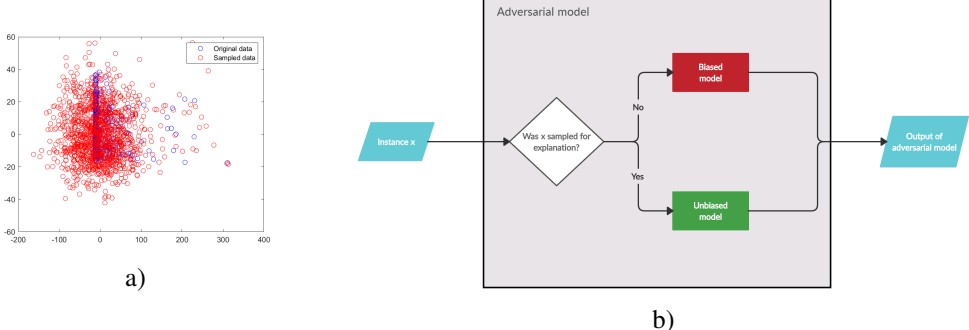

a)

b)

Figure 1: a) PCA based visualization of a part of the COMPAS dataset. The blue points show the original instances, and the red points represent instances generated with the perturbation sampling used in the LIME method. The distributions are notably different. b) The idea of the attack on explanation methods based on the difference of distributions. The attacker's adversarial model contains both the biased and unbiased model. The decision function that is part of the cheating model decides if the instance is outside the original distribution (i.e. used only for explanation) or an actual instance. If the case of an actual instance, the result of the adversarial model is equal to the result of the biased model, otherwise it is equal to the result of the unbiased model.

owners of prediction models could detect when their models are examined and return unbiased predictions in this case and biased predictions in normal use. This could have serious consequences in areas where predictive models' reliability and fairness are essential, e.g., in healthcare or banking. Such weaknesses can undermine users' trust in machine learning models in general and slow down technological progress.

In this work, we propose to change the main perturbation-based explanation methods and make them more resistant to manipulation attempts. In our solution, the problematic perturbation-based sampling is replaced with more advanced sampling, which uses modern data generators that better capture the distribution of the training dataset. We test three generators, the RBF network based generator (Robnik-Šikonja, 2016), random forest-based generator, available in R library semiArtificial (Robnik-Šikonja, 2019), as well as the generator using variational autoencoders (Miok et al., 2019). We show that the modified gLIME and gSHAP methods are much more robust than their original versions. For the IME method, which previously was not analyzed, we show that it is already quite robust. We release the modified explanation methods under the open-source license[1].

In this work, we use the term robustness of the explanation method as a notion of resilience against adversarial attacks, i.e. as the ability of an explanation method to recognize the biased classifier in an adversary environment. This type of robustness could be more formally defined as the number of instances where the adversarial model's bias was correctly recognized. We focus on the robustness concerning the attacks described in Slack et al. (2020). There are other notions of robustness in explanation methods; e.g., (Alvarez-Melis & Jaakkola, 2018) define the robustness of the explanations in the sense that similar inputs should give rise to similar explanations.

The remainder of the paper is organized as follows. In Section 2, we present the necessary background and related work on explanation methods, attacks on them, and data generators. In Section 3, we propose a defense against the described weaknesses of explanation methods, and in Section 4, we empirically evaluate the proposed solution. In Section 5, we draw conclusions and present ideas for further work.

## 2 BACKGROUND AND RELATED WORK

In this section, we first briefly describe the background on post-hoc explanation methods and attacks on them, followed by data generators and related works on the robustness of explanation methods.

### 2.1 POST-HOC EXPLANATION METHODS

The current state-of-the-art perturbation-based explanation methods, IME, LIME, and SHAP, explain predictions for individual instances. To form an explanation of a given instance, they measure the difference in prediction between the original instance and its neighboring instances, obtained

---

[1]https://anonymous.4open.science/r/5d550c62-5c5c-4ee3-81ef-ab96fe0838ca/

with perturbation sampling. Using the generated instances, the LIME method builds a local interpretable model, e.g., a linear model. The SHAP and IME methods determine the impact of the features as Shapley values from the coalitional game theory (Shapley, 1988). In this way, they assure that the produced explanations obey the four Shapley fairness axioms (Štrumbelj & Kononenko, 2013). Due to the exponential time complexity of Shapley value calculation, both methods try to approximate them. The three methods are explained in detail in the above references, and a formal overview is presented in Appendix A, while below, we present a brief description. In our exposition of the explanation methods, we denote with $f$ the predictive model and with $x$ the instance we are explaining.

Explanations of instances with the LIME method is obtained with an interpretable model $g$. The model $g$ has to be both locally accurate (so that it can obtain correct feature contributions) and simple (so that it is interpretable). Therefore in LIME, $g$ is a linear regression model trained on the instances sampled in the neighborhood of the explained instance $x$, which is first normalized. The perturbed explanation instances are generated by adding Gaussian noise to each feature of $x$ independently. Using linear models to determine the contributions of features, LIME is fast but may produce poor explanations for instances close to complex decision boundaries.

SHAP estimates Shapley values using the approach of LIME (i.e. local linear models computed with the weighted least squares algorithm) but using a different neighborhood and loss function for local accuracy, which ensures that Shapley values are obtained. The sampling of explanation instances in SHAP relies on the distribution set $D$, obtained from the training set, by computing the centroids of k-means clusters.

IME computes Shapley values using a pure sampling approach (no proxy with an interpretable model). Its sampling replaces part of the explained instance with existing values of selected features and then averages differences in the prediction of explained instance and samples. This makes the method slower but also potentially more robust as the method does not assume the shape of the dependency.

## 2.2 ATTACKS ON EXPLANATIONS

The idea of the attack on explanations presented in Slack et al. (2020) is that the attacker creates an adversarial (biased) model whose behavior it wants to hide from explanation methods, e.g., a racist model that does not grant credits to blacks. As Figure 1b illustrates, a part of the attacker's adversarial model is an unbiased model, which, e.g., does not take disputed attributes such as race into account. The adversarial model tries to manipulate the explanation method by behaving like an unbiased model on perturbed instances used for the explanation – in this case, it uses the unbiased model's output. On instances from the original distributions that are a part of normal model use, the adversarial model uses the biased model's output. To distinguish between the original and perturbation-sampled data, the adversarial model contains a decision model that selects the right type of model (biased or unbiased) based on the predicted type of instances.

The reason that the decision function can distinguish between the original data and sampled instances are the shortcomings of explanation methods that use the perturbation sampling (see Figure 1a). For that graph, we created new instances based on the COMPAS dataset (see Section 4) with the Gaussian noise as in the LIME explanation method. We mapped the original and sampled instances to 2D space using the PCA transformation. We can see that the sampled data is distributed differently than the original data. As the difference is revealed with PCA, other machine learning algorithms can also effectively distinguish between sampled and original data.

The attack can be formally defined in the following way. The attacker has a biased model $b$ and a sample $S$ from the distribution $X_{dist}$. The attacker has no additional information about the distribution $X_{dist}$. The attacker's goal is to produce an adversarial model $e$ that behaves like $b$ on the data from $X_{dist}$ but does not reveal $b$'s bias to the explanation methods. We assume that the attacker also has an unbiased model $\psi$ that hides the $f$'s bias. The attacker creates a decision model $d$, that should output 1 if the input instance $x$ is from $X_{dist}$ and 0 otherwise. The model $d$ is trained on $S$ and generated perturbation samples. The attacker creates the adversarial model $e$ with its output defined by the following equation:

$$e(x) = \begin{cases} b(x), & d(x) = 1 \\ \psi(x), & d(x) = 0 \end{cases} \tag{1}$$

## 2.3 DATA GENERATORS

We used three different data generators based on different algorithms, modeling the distribution of the training set: variational autoencoder with Monte Carlo dropout (Miok et al., 2019), RBF network (Robnik-Šikonja, 2016), and random forest ensemble (Robnik-Šikonja, 2019). In the remainder of the paper, we refer to the listed generators consecutively as MCD-VAE, rbfDataGen, and TreeEnsemble.

Autoencoder (AE) consists of two neural networks called encoder and decoder. It aims to compress the input instances by passing them through the encoder and then reconstructing them to the original values with the decoder. Once the AE is trained, it can be used to generate new instances. Variational autoencoder (Doersch, 2016) is a special type of autoencoder, where the vectors $z$ in the latent dimension (output of the encoder and input of the decoder) are normally distributed. Encoder is therefore approximating the posterior distribution $p(z|x)$, where we assume $p(z|x) \sim \mathcal{N}(\mu_x, \Sigma_x)$. The generator proposed by Miok et al. (2019) uses the Monte Carlo dropout (Gal & Ghahramani, 2016) on the trained decoder. The idea of this generator is to propagate the instance $x$ through the encoder to obtain its latent encoding $z$. This can be propagated many times through the decoder, obtaining every time a different result due to the Monte Carlo dropout but preserving similarity to the original instance $x$.

The RBF network (Moody & Darken, 1989) uses Gaussian kernels as hidden layer units in a neural network. Once the network's parameters are learned, the rbfDataGen generator (Robnik-Šikonja, 2016) can sample from the normal distributions, defined with obtained Gaussian kernels, to generate new instances.

The TreeEnsemble generator (Robnik-Šikonja, 2019) builds a set of random trees (forest) that describe the data. When generating new instances, the generator traverses from the root to the leaves of a randomly chosen tree, setting values of features in the decision nodes on the way. When reaching a leaf, it assumes that it has captured the dependencies between features. Therefore, the remaining features can be generated independently according to the observed empirical distribution in this leaf. For each generated instance, all attributes can be generated in one leaf, or another tree can be randomly selected where unassigned feature values are filled in. By selecting different trees, different features are set in the interior nodes and leaves.

## 2.4 RELATED WORK ON ROBUSTNESS OF EXPLANATIONS

The adversarial attacks on perturbation based explanation methods were proposed by Slack et al. (2020), who show that LIME and SHAP are vulnerable due to the perturbation based sampling used. We propose the solution to the exposed weakness in SHAP and IME based on better sampling using data generators adapted to the training set distribution.

In general, the robustness of explanation methods has been so far poorly researched. There are claims that post-hoc explanation methods shall not be blindly trusted, as they can mislead users (deliberately or not) and disguise gender and racial discrimination (Lipton, 2016). Selbst & Barocas (2018) and Kroll et al. (2017) showed that even if a model is completely transparent, it is hard to detect and prevent bias due to the existence of correlated variables.

Specifically, for deep neural networks and images, there exist adversarial attacks on saliency map based interpretation of predictions, which can hide the model's bias (Dombrowski et al., 2019; Heo et al., 2019; Ghorbani et al., 2019). Dimanov et al. (2020) showed that a bias of a neural network could be hidden from post-hoc explanation methods by training a modified classifier that has similar performance to the original one, but the importance of the chosen feature is significantly lower.

The kNN-based explanation method, proposed by Chakraborty et al. (2020), tries to overcomes the inadequate perturbation based sampling used in explanation methods by finding similar instances to the explained one in the training set instead of generating new samples. This solution is inadequate for realistic problems as the problem space is not dense enough to get reliable explanations. Our defense of current post-hoc methods is based on superior sampling, which has not yet been tried. Saito et al. (2020) use the neural CT-GAN model to generate more realistic samples for LIME and prevent the attacks described in Slack et al. (2020). We are not aware of any other defenses against the adversarial attacks on post-hoc explanations.

## 3 ROBUSTNESS THROUGH BETTER SAMPLING

We propose the defense against the adversarial attacks on explanation methods that replaces the problematic perturbation sampling with a better one, thereby making the explanation methods more robust. We want to generate the explanation data in such a way that the attacker cannot determine whether an instance is sampled or obtained from the original data. With an improved sampling, the adversarial model shown in Figure 1b shall not determine whether the instance $x$ was generated by the explanation method, or it is the original instance the model has to label. With a better data generator, the adversarial model cannot adjust its output properly and the deception, described in Section 2.2, becomes ineffective.

The reason for the described weakness of LIME and SHAP is inadequate sampling used in these methods. Recall that LIME samples new instances by adding Gaussian noise to the normalized feature values. SHAP samples new instances from clusters obtained in advance with the k-means algorithm from the training data.

Instead of using the Gaussian noise with LIME, we generate explanation samples for each instance with one of the three better data generators, MCD-VAE, rbfDataGen, or TreeEnsemble (see Section 2.3). We call the improved explanation methods gLIME and gSHAP (g stands for generator-based). Using better generators in the explanation methods, the decision function in the adversarial model will less likely determine which predicted instances are original and which are generated for the explanation purposes.

Concerning gLIME, we generate data in the vicinity of the given instance using MCD-VAE, as the LIME method builds a local model. Using the TreeEnsemble and rbfDataGen generators, we do not generate data in the neighborhood of the given instance but leave it to the proximity measure of the LIME method to give higher weights to instances closer to the explained one.

In SHAP, the perturbation sampling replaces the values of hidden features in the explained instance with the values from the distribution set $D$. The problem with this approach is that it ignores the dependencies between features. For example, in a simple dataset with two features, house size, and house price, let us assume that we hide the house price, but not the house size. These two features are not independent because the price of a house increases with its size. Suppose we are explaining an instance that represents a large house. Disregarding the dependency, using the sampled set $D$, SHAP creates several instances with a low price, as such instances appeared in the training set. In this way, the sampled set contains many instances with a small price assigned to a large house, from which the attacker can determine that these instances were created in perturbation sampling and serve only for the explanation.

In the proposed gSHAP, using the MCD-VAE and TreeEnsemble generators, the distribution set $D$ is generated in the vicinity of the explained instance. In the sampled instances, some feature values of the explained instance are replaced with the generated values, but the well-informed generators consider dependencies between features detected in the original distribution. This will make the distribution of the generated instances very similar to the original distribution. In our example, the proposed approach generates new instances around the instance representing a large house, and most of these houses will be large. As the trained generators capture the original dataset's dependencies, these instances will also have higher prices. This will make it difficult for the attacker to recognize the generated instances used in explanations. The advantage of generating the distribution set close to the observed instance is demonstrated in Appendix B.

The rbfDataGen generator does not support the generation of instances around a given instance. Therefore, we generate the sampled set based on the whole training set and not for each instance separately (we have this option also for TreeEnsemble). This is worse than generating the distribution set around the explained instance but still better than generating it using the k-means sampling in SHAP. There are at least three advantages. First, the generated distribution set $D$ can be larger. The size of the k-means distribution set cannot be arbitrarily large because it is limited by the number of clusters in the training set. Second, the centroids of the clusters obtained by the k-means algorithm do not necessarily provide a good summary of the training set and may not be valid instances from training set distribution. They also do not capture well the density of instances (e.g., most of the data from the training set could be grouped in one cluster). Third, using the proposed generators, SHAP becomes easier to use compared to the k-means generator, where users have to determine

| Data set | # inst. | # features | # categorical | sensitive | unrelated 1 | unrelated 2 |
|----------|---------|------------|---------------|-----------|-------------|-------------|
| COMPAS   | 6172    | 7          | 4             | *race*    | *random1*   | *random2*   |
| German   | 1000    | 25         | 15            | *gender*  | *pctOfIncome* | */*       |
| CC       | 1994    | 100        | 0             | *racePctWhite* | *random1* | *random2* |

Table 1: Basic information and sensitive features in the the used data sets. The target variable is not included in the number of features and is binary for all data sets. The *pctOfIncome* full name is *loanRateAsPercentOfIncome*.

the number of clusters, while the data generators we tested can be used in the default mode without parameters,

## 4  EVALUATION

To evaluate the proposed improvements in the explanation methods, we first present the used datasets in Section 4.1, followed by the experiments. In Section 4.2, we test the robustness of gLIME, gSHAP, and gIME against the adversarial models. To be more realistic, we equipped the adversarial models with the same improved data generators we used in the explanation methods. It is reasonable to assume that attackers could also use better data generators when preparing their decision function, making their attacks much stronger. As the evaluation metric for the success of deception, we use the proportion of instances where the adversarial model deceived the explanation methods so that they did not detect sensitive attributes as being important in the prediction models. In Section 4.3, we test if enhanced generators produce different explanations than the original ones. As the attacker might decide to employ deception only when it is really certain that the predicted instance is used inside the explanation method, we test different thresholds of the decision function $d$ from Equation (1) (currently set to $0.5$). We report on this analysis in Section 4.4.

### 4.1  SENSITIVE DATASETS PRONE TO DECEPTION

Following (Slack et al., 2020), we conducted our experiments on three data sets from domains where a biased classifier could pose a critical problem, such as granting a credit, predicting crime recidivism, and predicting the violent crime rate. The basic information on the data sets is presented in Table 1. The statistics were collected after removing the missing values from the data sets and before we encoded categorical features as one-hot-encoded vectors.

COMPAS (Correctional Offender Management Profiling for Alternative Sanctions) is a risk assessment used by the courts in some US states to determine the crime recurrence risk of a defendant. The dataset (Angwin et al., 2016)) includes criminal history, time in prison, demographic data (age, gender, race), and COMPAS risk assessment of the individual. The dataset contains data of 6,172 defendants from Broward Couty, Florida. The sensitive attribute in this dataset (the one on which the adversarial model will be biased) is race. African Americans, whom biased model associates with a high risk of recidivism, represent $51.4\%$ of instances from the data set. This set's target variable is the COMPAS score, which is divided into two classes: a high and low risk. The majority class is the high risk, which represents $81.4\%$ of the instances.

The German Credit dataset (German for the rest of the paper) from the UCI repository (Dua & Graff, 2019) includes financial (bank account information, loan history, loan application information, etc.) and demographic data (gender, age, marital status, etc.) for 1,000 loan applicants. A sensitive attribute in this data set is gender. Men, whom the biased model associates with a low-risk, represent $69\%$ of instances. The target variable is the loan risk assessment, divided into two classes: a good and a bad customer. The majority class is a good customer, which represents $70\%$ of instances.

Communities and Crime (CC) data set (Redmond & Baveja, 2002) contains data about the rate of violent crime in US communities in 1994. Each instance represents one community. The features are numerical and represent the percentage of the community's population with a certain property or the average of population in the community. Features include socio-economic (e.g., education, house size, etc.) and demographic (race, age) data. The sensitive attribute is the percentage of the white race. The biased model links instances where whites' percentage is above average to a low rate of violent crime. The target variable is the rate of violent crime divided into two classes: high and low. Both classes are equally represented in the data set.

## 4.2 ROBUSTNESS OF EXPLANATION METHODS

To evaluate the robustness of explanation methods with added generators (i.e. gLIME, gSHAP, and gIME), we split the data into training and evaluation set in the ratio $90\% : 10\%$. We used the same training set for the training of adversarial models and explanation methods. We encoded categorical features as one-hot-encoded vectors.

We simulated the adversarial attack for every combination of generators used in explanation methods and adversarial models (except for method IME, where we did not use rbfDataGen, which cannot generate instances in the neighborhood of a given instance). When testing SHAP, we used two variants of the TreeEnsemble generator for explanation: generating new instances around the explained instance and generating new instances according to the whole training set distribution. In the LIME testing, we used only the whole training set variant of TreeEnsemble inside the explanation. In the IME testing, we only used the variant of Tree Ensemble that fills in the hidden values (called TEnsFillIn variant). For training the adversarial models, we used the whole training set based variant of TreeEnsemble, except for IME, where we used the TEnsFillIn variant. These choices reflect the capabilities of different explanation method and attempt to make both defense and attack realistic (as strong as possible). More details on training the decision model $d$ inside the adversarial model can be found in Appendix D.

In all cases, the biased model $b$ (see Section 2.2) was a simple function depending only on the value of the sensitive feature. The unbiased model $\psi$ depended only on the values of unrelated features. The sensitive and unrelated features are shown on the right-hand side of Table 1. Features *random1* and *random2* were uniformly randomly generated from the $\{0, 1\}$ set. On COMPAS and CC, we simulated two attacks, one with a single unrelated feature (the result of $\psi$ depends only on the value of the unrelated feature 1), and another one with two unrelated features.

For every instance in the evaluation set, we recorded the most important feature according to the used explanation method (i.e. the feature with the largest absolute value of the contribution as determined by the explanation method). The results are shown as a heatmap in Figure 2. The green color means that the explanation method was deceived in less than $30\%$ of cases (i.e. the sensitive feature was recognized as the most important one in more than 70 % of the cases as the scale on the heatmap suggests), and the red means that it was deceived in more than $70\%$ of cases (the sensitive feature was recognized as the most important one in less than 30 % of the cases as the scale on the heatmap suggests). We consider deception successful if the sensitive feature was not recognized as the most important by the explanation method (the sensitive features are the only relevant features in biased models $b$).

The gLIME method is more robust with the addition of rbfDataGen and TreeEnsemble than LIME and less robust with the addition of MCD-VAE. This suggests that parameters for MCD-VAE were not well-chosen, and this pattern repeats for other explanation methods. Both TreeEnsemble and rbfDataGen make gLIME considerably more robust on COMPAS and German datasets, but not on the CC dataset. We believe the reason for that is that features in CC are strongly interdependent, and many represent the same attribute as a fraction of value, e.g., we have the share of the white population, the share of the Asian population, and so on. This interdependence dictates that all fractions have to add up to $1$. The data generators are unlikely to capture such strong conditional dependencies, but the adversarial model's decision function is likely to approximate it.

The gSHAP method is most robust when using the TreeEnsemble generator, but it shows less robust behavior than the gLIME method. The reason for that could be in the feature value replacement strategy used by the SHAP and gSHAP methods, which change only some of the feature's values with the help of the distribution set. This can lead to out-of-distribution instances if the instances in the distribution set are not close enough to the explained instance. With gLIME, we generate complete instances that are more likely to be in the distribution of the training set.

IME is quite robust even with the perturbation sampling, as we expected. This suggests that IME is the most robust of all three tested explanation methods and shall be considered the chosen method in sensitive situations. The gIME results when using the TreeEnsemble generator (TEnsFillIn variant) are comparable to the original IME variant results. This suggests that sampling from a smaller data set, which represents the neighborhood of the explained instance, does not decrease the method's robustness.

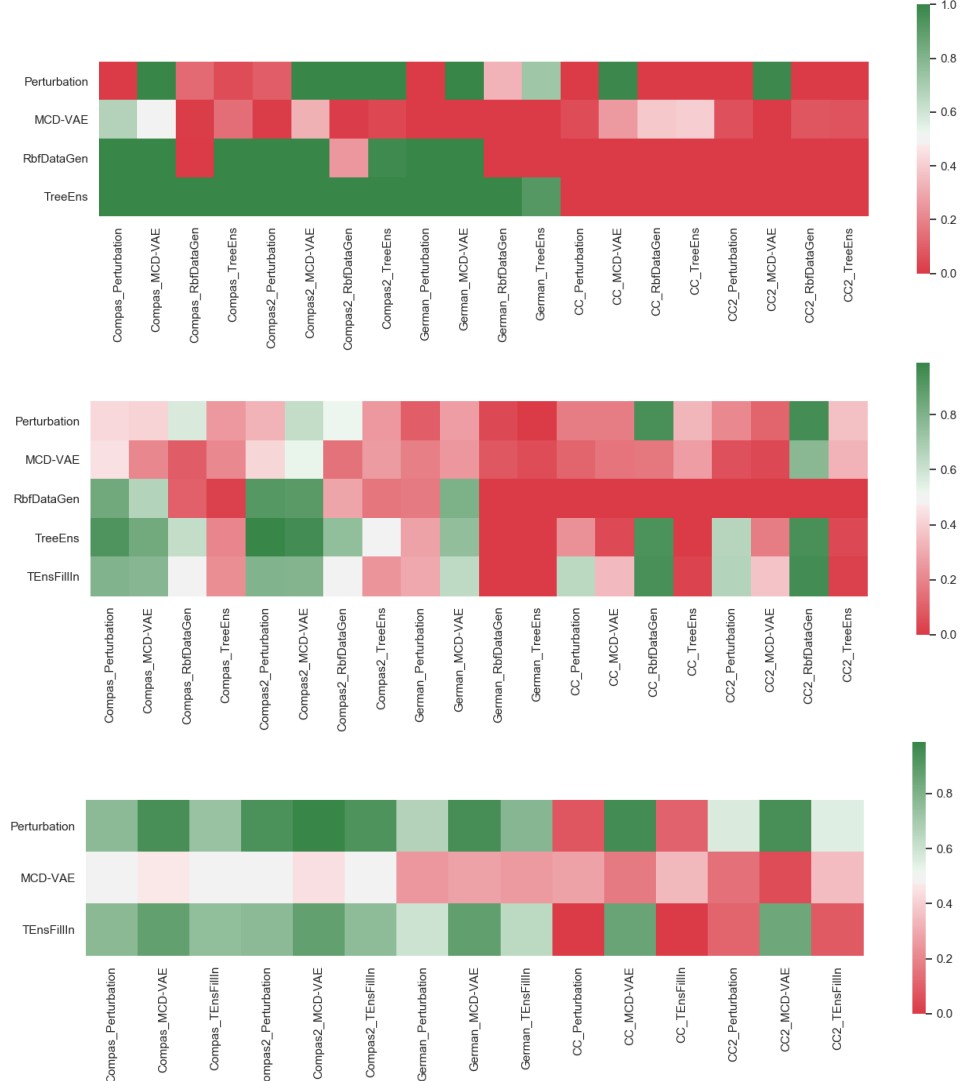

Figure 2: The robustness results for gLIME (top), gSHAP (middle), and gIME (bottom). The graphs show the proportion of evaluation set instances, where the sensitive feature was recognized as the most important by the used explanation method. Rows represent the generators used for explanations. The column labels consist of the name of the data set on which the experiment was performed and the name of the generator used for training of the adversarial model. Compas2 and CC2 denote an attack with two independent features. Perturbation represents the original sampling used in LIME, SHAP, and IME, TEnsFillIn represents the TreeEnsemble variant where new instances are generated around the given one, and TreeEns represents the generation from the whole distribution.

## 4.3 COMPARING EXPLANATIONS OF ORIGINAL AND MODIFIED METHODS

We checked if improved data generators affect explanations in a non-adversary environment. We compared the explanations of original and modified methods for four different classification models on three different datasets. As a measure of differences, we use the mean of the absolute difference between the original and enhanced explanations over the whole evaluation set. Details of these experiments are reported in Appendix F, below we shortly summarize the results, contained in Table 4.

The differences between original LIME and gLIME explanations are considerable (see the top part of Table 4). This is not surprising since LIME fits local linear models in its local explanations, which can strongly differ even for small perturbations of the model's input. The differences in SHAP and IME explanations are small (the average MAD is almost negligible). We can conclude that explanations of gSHAP and gIME are not significantly different from SHAP and IME in the non-adversary environment.

### 4.4 BEHAVIOR OF ADVERSARIAL CLASSIFIER WITH DIFFERENT PREDICTION THRESHOLDS

The attacker might be wary of being discovered by the regulator and decide to employ deception only when it is really certain that the predicted instance is used inside the explanation method. We express different levels of attacker's conservatism by changing the decision function threshold $d$ from Equation (1), where it is currently set to $0.5$. We tested the robustness of modified explanation methods on the COMPAS dataset with adversarial models that use different certainty thresholds for deploying the biased classifier (simulating different levels of aggressiveness). For results and a more detailed description of the experiment, see Appendix G. Below, we shortly summarize the results.

Even with different thresholds, gIME is still the most robust of the three explanation methods, and treeEnsemble still gives the best results as the data generator. The sensitive feature is recognized as the most important more often when the threshold is lower, which is expected as in this case the adversarial model behaves more aggressively. While the percentage of the instances on which the racist behavior of the adversarial model is recognized drops with higher values of the thresholds, it remains high enough to warn the regulator about the problematic behavior of the prediction model (especially in the case of gSHAP and gIME using treeEnsemble as data generator). We can conclude that the enhanced explanation methods remain robust enough, even with more conservative adversarial models.

## 5 CONCLUSIONS

We presented the defense against adversarial attacks on explanation methods. The attacks exploit the shortcomings of perturbation sampling in post-hoc explanation methods. This sampling used in these methods produces instances too different from the original distribution of the training set. This allows unethical owners of biased prediction models to detect which instances are meant for explanation and label them in an unbiased way. We replaced the perturbation sampling with data generators that better capture the distribution of a given data set. This prevents the detection of instances used in explanation and disarms attackers. We have shown that the modified gLIME and gSHAP explanation methods, which use better data generators, are more robust than the original variants, while IME is already quite robust. The difference in explanation values between original and enhanced gSHAP and gIME is negligible, while for gLIME, it is considerable. Our preliminary results in Appendix C show that using the TreeEnsemble generator, the gIME method converges faster and requires from 30-50% fewer samples.

The effectiveness of the proposed defense depends on the choice of the data generator and its parameters. While the TreeEnsemble generator turned out the most effective in our evaluation, in practice, several variants might need to be tested to get a robust explanation method. Inspecting authorities shall be aware of the need for good data generators and make access to training data of sensitive prediction models a legal requirement. Luckily, even a few non-deceived instances would be enough to raise the alarm about unethical models.

This work opens a range of possibilities for further research. The proposed defense and attacks shall be tested on other data sets with a different number and types of features, with missing data, and on different types of problems such as text, images, and graphs. The work on useful generators shall be extended to find time-efficient generators with easy to set parameters and the ability to generate new instances in the vicinity of a given one. Generative adversarial networks (Goodfellow et al., 2014) may be a promising line of research. The TreeEnsemble generator, currently written in pure R, shall be rewritten in a more efficient programming language. The MCD-VAE generator shall be studied to allow automatic selection of reasonable parameters for a given dataset. In SHAP sampling, we could first fix the values of the features we want to keep, and the values of the others would be generated using the TreeEnsemble generator.

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

## A  DETAILS ON POST-HOC EXPLANATION METHODS

For the sake of completeness, we present further details on the explanation methods LIME (Ribeiro et al., 2016), SHAP (Lundberg & Lee, 2017), and IME (Štrumbelj & Kononenko, 2013). Their complete description can be found in the above-stated references. In our exposition of the explanation methods, we denote with $f$ the predictive model, with $x$ the instance we are explaining, and with $n$ the number of features describing $x$.

### A.1  LIME

Explanations of instances with the LIME method is obtained with an interpretable model $g$. The model $g$ has to be both locally accurate (so that it can obtain correct feature contributions) and simple (so that it is interpretable).

The explanation of the instance $x$ for the predictive model $f$, obtained with the LIME method, is defined with the following equation:

$$\xi(x) = \underset{g \in G}{\arg\min}(\mathcal{L}(f, g, \pi_x) + \Omega(g)), \tag{2}$$

where $\Omega(g)$ denotes the measure of complexity for interpretable model, $\pi_x(z)$ denotes the proximity measure between $x$ and generated instances $z$, $\mathcal{L}(f, g, \pi_x)$ denotes the measure of local fidelity of interpretable model $g$ to the prediction model $f$, and $G$ denotes the set of interpretable models. We use the linear version of the LIME method, where $G$ represents the set of linear models. With $x'$ we denote the normalized presentation of instance $x$, i.e. the numerical attributes have their mean set to 0 and variance to 1, and the categorical attributes contain value 1 if they have the same value

as the exoplained instance and 0 otherwise. The proximity function $\pi_x$ is defined on the normalized instances (hence we use the notation $\pi_{x'}$), and uses the exponential kernel: $\pi_{x'}(z') = e^{\frac{d(x',z')}{\sigma^2}}$, where $d(x', z')$ denotes a distance measure between $x'$ and $z'$. The local fidelity measure $\mathcal{L}$ from Equation (2) is defined as:

$$\mathcal{L}(f, g, \pi_{x'}) = \sum_{z' \in \mathcal{Z}} \pi_{x'}(z')(f(z) - g(z'))^2, \tag{3}$$

where $\mathcal{Z}$ denotes the set of samples. In LIME, each generated sample is obtained by adding Gaussian noise to each feature of $x'$ independently.

Using linear models as the set of interpretable models, LIME is relatively fast but may produce poor explanations for instances close to complex decision boundaries.

## A.2 SHAP

We refer to SHAP as the method called Kernel SHAP by (Lundberg & Lee, 2017). SHAP essentially estimates Shapley values using LIME's approach, which means that the calculation of explanations is fast due to the use of local linear models computed with the weighted least squares algorithm. The explanation of the instance $x$ with the SHAP method are feature contributions $\phi_i$, $i = 1, 2, ..., n$ that are coefficients of the linear model $g(x') = \phi_0 + \sum_{i=1}^{n} \phi_i \cdot x'_i$, obtained with LIME, where $x'_i \in \{0, 1\}$ for $i \in \{1, 2, ..., n\}$.

As LIME does not compute Shapley values, this property is assured with the proper selection of functions $\Omega(g), \pi_{x'}(z')$, and $\mathcal{L}(f, g, \pi_{x'})$. Let us first define the function $h_x(z')$ that maps from the $\{0, 1\}^n$ to the original feature space. The function $h_x$ is defined implicitly with the equation $f(h_x(z')) = E[f(z)|z_i = x_i \, \forall i \in \{j; z'_j = 1\}]$. This is the expected value of the model $f$ when we do not know the values of features at indices where $z'$ equals 0 (these values are hidden). Functions $\Omega(g), \pi_{x'}(z')$ and $\mathcal{L}(f, g, \pi_{x'})$ that enforce the computation of Shapley values are:

$$\Omega(g) = 0,$$

$$\pi_{x'}(z') = \frac{n}{\binom{n}{|z'|} \cdot |z'| \cdot (n - |z'|)},$$

$$\mathcal{L}(f, g, \pi_{x'}) = \sum_{z' \in \mathcal{Z}} (f(h_x(z')) - g(z'))^2 \cdot \pi_{x'}(z'),$$

where $|z'|$ denotes the number of nonzero features of $z'$ and $\mathcal{Z} \subseteq 2^{\{0,1\}^n}$.

The main purpose of the sampling in this method is to determine the value $f(h_x(z'))$ because in general predictive models cannot work with hidden values. To determine $f(h_x(z'))$, SHAP uses the distribution set $D$ which we obtain from the training set. For $D$, SHAP takes the centroids of clusters obtained by the k-means clustering algorithm on the training set. The number of clusters is set by the user. Value of $f(h_x(z'))$ is determined by the following sampling:

$$f(h_x(z')) = E[f(z)|z_i = x_i \, \forall i \in \{j; z'_j = 1\}] = \frac{1}{|D|} \sum_{d \in D} f(x_{[x_i = d_i, z'_i = 0]}), \tag{4}$$

where $x_{[x_i = d_i, z'_i = 0]}$ denotes instance $x$ with features that are 0 in $z'$ being set to the feature values from $d$.

## A.3 IME

The explanation of $x$ with the method IME are feature contributions $\phi_i$, $i = 1, 2, ..., n$. Štrumbelj & Kononenko (2013) shoved that Shapley values are the only solution that takes into account contributions of interactions for every subset of features in a fair way. The $i$-th feature contribution can be calculated with the following expresiion:

$$\phi_i(x) = \frac{1}{n} \sum_{\pi \in S_n} \sum_{w \in \mathcal{X}} p(w) \cdot (f(w_{[w_i = x_i, i \in Pre^i(\pi) \cup \{i\}]}) - f(w_{[w_i = x_i, i \in Pre^i(\pi)]})), \tag{5}$$

where $S_n$ denotes a group of permutations of $n$ elements, $\mathcal{X}$ denotes the training set instances, $Pre^i(\pi)$ represents the set of indices that precedes $i$ in the permutation $\pi$, i.e. $Pre^i(\pi) = \{j; \pi(j) < \pi(i)\})$. Let $p(w)$ denote the probability of the instance $w$ in $\mathcal{X}$ and let $w_{[formula]}$ denote the instance $w$ with some of its features values changed according to the *formula*. To calculate $\phi_i(x)$, we have to go through $|\mathcal{X}| \cdot n!$ iterations, which can be slow. Therefore, the method IME uses the following sampling. The sampling population of $i$-th feature is $V_{\pi,w} = (f(w_{[w_i=x_i, i \in Pre^i(\pi) \cup \{i\}]}) - f(w_{[w_i=x_i, i \in Pre^i(\pi)]}))$ for every combination of the permutation $\pi$ and instance $w$. IME draws $m_i$ samples $V_1, ..., V_{m_i}$ at random with repetition. The estimate of $\phi_i(x)$ is defined with the equation:

$$\hat{\phi}_i = \frac{1}{m_i} \sum_{i=1}^{m_i} V_i. \tag{6}$$

Contrary to SHAP, IME does not use approximation with linear models, which compute all features' contributions at once but has to compute the Shapley values by averaging over a large enough sample for each feature separately. This makes the method slower but also potentially more robust as the method does not assume the shape of the dependency in the space of normalized features.

## B  DEMONSTRATION OF BETTER SAMPLING IN SHAP

The graphs in Figure 3 show the PCA based 2D space of the evaluation part of the COMPAS dataset (see Section 4 for the dataset description). The left-hand side shows the SHAP-generated sampled instances using the k-means algorithm (14 clusters determined by the silhouette score (Rousseeuw (1987)). The sample produced with the MCD-VAE generator in gSHAP is shown on the right-hand side. This sample is much more similar to the original distribution compared to the SHAP sampling.

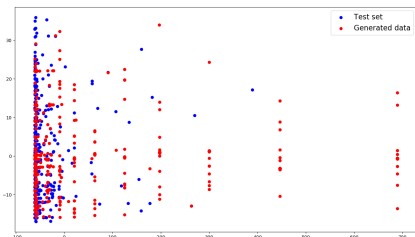 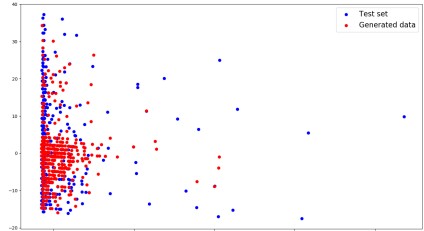

Figure 3: Visual comparison of original and sampled distributions for the COMPASS dataset. The SHAP k-means based generator (left) produces instances less similar to the original data, compared to the MCD-VAE generator (right).

## C  IMPROVED IME CONVERGENCE RATE WITH THE TREEENSEMBLE GENERATOR

Preliminary, we tested how better generators affect the convergence rate of the IME explanation method. The preliminary results in Table 2 show a significant reduction in the number of needed samples and a slight increase in the error compared to the original perturbation sampling. Note that the error measure deployed is biased in favor of the perturbation sampling, which was used to determine the gold standard. This was determined with the sampling population's variance, as described in Štrumbelj & Kononenko (2010).

## D  TRAINING DISCRIMINATOR FUNCTION OF THE ATTACKER

Details of training attacker's decision models $d$ (see Figure 1b) is described in Algorithms 1, 2, and 3. We used a slightly different algorithm for each of the three explanation methods, LIME, SHAP,

**COMPAS dataset**

| Classifier | Error | | # samples | | | |
| | Perturb. | TEnsFillIn | Perturb. | TEnsFillIn | Reduction % | CA % |
|---|---|---|---|---|---|---|
| Naive Bayes | 0.0076 | 0.0217 | 18571 | 11278 | 39 | 83 |
| Linear SVM | 0.0033 | 0.0080 | 8244 | 4423 | 46 | 84 |
| Random forest | 0.0049 | 0.0221 | 45372 | 26960 | 41 | 80 |
| Neural network | 0.0057 | 0.0130 | 16157 | 8841 | 45 | 84 |

**German dataset**

| Classifier | Error | | # samples | | | |
| | Perturb. | TEnsFillIn | Perturb. | TEnsFillIn | Reduction % | CA % |
|---|---|---|---|---|---|---|
| Naive Bayes | 0.0076 | 0.0201 | 56052 | 39257 | 30 | 77 |
| Linear SVM | 0.0005 | 0.0013 | 3877 | 2157 | 44 | 69 |
| Random forest | 0.0046 | 0.0141 | 92478 | 66639 | 28 | 74 |
| Neural network | 0 | 0 | 0 | 26 | / | 69 |

**CC dataset**

| Classifier | Error | | # samples | | | |
| | Perturb. | TEnsFillIn | Perturb. | TEnsFillIn | Reduction % | CA % |
|---|---|---|---|---|---|---|
| Naive Bayes | 0.0028 | 0.0046 | 32910 | 20117 | 39 | 70 |
| Linear SVM | 0.0009 | 0.0048 | 73324 | 39098 | 47 | 62 |
| Random forest | 0.0032 | 0.0045 | 109958 | 58852 | 46 | 79 |
| Neural network | 0.0012 | 0.0061 | 144183 | 70020 | 51 | 72 |

Table 2: Comparison of original perturbation sampling and the TreeEnsemble generator with data fill-in inside the IME method. The results show average scores for the evaluation set. The column Perturb. presents the perturbation based sampling and TEnsFillIn presents the sampling using TreeEnsemble generator with missing parts of instances filled in. The Reduction column shows the reduction in the number of samples using the TEnsFillIn method compared to perturbations. The CA stands for the classification accuracy of the explained classifier on the evaluation set.

and IME, as each method uses a different sampling. Algorithms first create out-of-distribution instances by method-specific sampling. The training sets for decision models are created by labeling the created instances with 0; the instances from sample $S$ (to which the attacker has access) from distribution $X_{dist}$ are labeled with 1. Finally, the machine learning model *dModel* is trained on this training set and returned as $d$. In our experiments, we used random forest classifier as *dModel* and the training part of each evaluation dataset as $S$.

## E    HEATMAPS AS TABLES

We present the information contained in Figure 2 in a more detailed tabular form in Table 3.

## F    COMPARING EXPLANATIONS OF ORIGINAL AND MODIFIED METHODS

We check if improved data generators affect explanations in non-adversary environment. We split the dataset into training and evaluation set in the ratio $90\% : 10\%$, and trained four classifiers from Python scikit-learn (Pedregosa et al. (2011)) library: Gaussian naive Bayes, linear SVC (SVM for classification), random forest, and neural network. We explained the predictions of each classifier on the evaluation set with every combination of explanation methods and generators used in the adversarial attack experiments. For instances in the evaluation set, we measured the mean absolute

---

**Algorithm 1:** Training of the decision model $d$, used by the attacker to distinguish between instances from distribution $X_{dist}$ and samples produced by explanation methods LIME or gLIME.

---

**Input:** $S = \{(x_i)_{i=1}^m\}$: training set, *nSamples*: number of generated instances for each instance $x_i \in D$, $gen$: data generator, *dModel*: machine learning algorithm
**Output:** Classifier $d$ that outputs 1 if its input $x$ is from $X_{dist}$ and 0 otherwise
$X \leftarrow \emptyset$ // Training set for *dModel*
$gen.fit(S)$ // Train the data generator on $S$
**for** $i = 1$ *to* $m$ **do**
    $X \leftarrow X \cup (x_i, 1)$ // Add an instance from distribution
    $G \leftarrow gen.newdata(nSamples, x_i)$ // Generate *nSamples* new samples around $x_i$
    **for** $j = 1$ *to* $nSamples$ **do** // Add *nSamples* out of distribution instances
        $X \leftarrow X \cup (G[j], 0)$ // Add $j$-th instance from set $G$ to $X$
    **end**
**end**
$d \leftarrow dModel.fit(X)$ // Fit model *dModel* to set $X$ and save it in $d$
**return** $d$

---

**Algorithm 2:** Training of the decision model $d$, used by attacker to distinguish between instances from distribution $X_{dist}$ and samples produced by explanation methods SHAP or gSHAP.

---

**Input:** $S = \{(x_i)_{i=1}^m\}$: training set, *nSamples*: number of generated instances for each instance $x_i \in D$, $k$: size of the generated distribution set, $gen$: data generator, *dModel*: machine learning algorithm
**Output:** Classifier $d$ that outputs 1 if its input $x$ is from $X_{dist}$ and 0 otherwise
$X \leftarrow \emptyset$ // Training set for *dModel*
$gen.fit(S)$ // Train the data generator on $S$
**if** $gen == KMeans$ *or* $gen == rbfDataGen$ *or* $gen == treeEnsemble$ **then**
    $D \leftarrow gen.newdata(k)$ // Generate the distribution set with KMeans, rbfDataGen or treeEnsemble
**end**
**for** $i = 1$ *to* $nSamples$ **do** // Add *nSamples* out of distribution instances
    $x \leftarrow$ random instance from $S$
    **if** $gen == MCD - VAE$ *or* $gen == treeEnsembleFill$ **then**
        $w \leftarrow gen.newdata(1, x)$ // Generate an instance $w$ in the vicinity of $x$
    **end**
    **else** // KMeans, treeEnsemble or rbfDataGen
        $w \leftarrow$ take a random instance from $D$
    **end**
    $M \leftarrow$ choose a random subset of $\{1, 2, ..., len(x)\}$ // Choose random features
    $x[M] \leftarrow w[M]$ // Replace the values of chosen features in $x$ with values from $w$ as in SHAP method
    $X \leftarrow X \cup (x, 0)$ // Add out of distribution instance
**end**
**for** $i = 1$ *to* $m$ **do** // Add instances from distribution
    $X \leftarrow X \cup (x_i, 1)$
**end**
$d \leftarrow dModel.fit(X)$ // Fit model *dModel* to set $X$ and save it in $d$
**return** $d$

---

difference (MAD) of modified explanation methods, defined with the following equation:

$$\text{MAD}_{gen}(x) = \frac{1}{n} \sum_{i=1}^{n} |\phi_i^{gen}(x) - \phi_i(x)|, \tag{7}$$

where $\phi_i^{gen}(x)$ and $\phi_i(x)$ represent the explanations of $i$-th feature returned by the modified and original explanation method, respectively (recall that $n$ denotes the number of features in the data set).

We experimented on three datasets. The COMPAS dataset is described in Section 4.1. In addition to that, we used synthetic dataset condInd from Robnik-Šikonja & Kononenko (2008), and Ionosphere

---

**Algorithm 3:** Training of the decision model $d$, used by attacker to distinguish between instances from distribution $X_{dist}$ and samples produced by explanation methods IME or gIME.

---

**Input:** $S = \{(x_i)_{i=1}^m\}$: training set, *nSamples*: number of generated instances for each instance
   $\quad\quad x_i \in D$, $gen$: data generator, *dModel*: machine learning algorithm
**Output:** Classifier $d$ that outputs 1 if its input $x$ is from $X_{dist}$ and 0 otherwise
$X \leftarrow \emptyset$ // Training set for *dModel*
$gen.fit(S)$ // Train the data generator on set $S$
**for** $i = 1$ *to* $m$ **do**
   $X \leftarrow X \cup (x_i, 1)$ // Add an instance from distribution
   **for** $j = 1$ *to* *nSamples* **do** // Add *nSamples* out of distribution instances
      $w \leftarrow gen.newdata(1, x_i)$ // Generate an instance $w$ in the vicinity of $x_i$
      $b_1 \leftarrow w$ // First out of distribution instance
      $b_2 \leftarrow w$ // Second out of distribution instance
      $\pi \leftarrow$ choose a random permutation from $S_{len(x_i)}$ // i.e. a random permutation of $x_i$'s
        features
      $idx \leftarrow$ choose a random number from $\{1, 2, ..., len(x_i)\}$
        $M_1 \leftarrow \{k \in \{1, 2, ..., len(x_i)\}, \pi(k) < \pi(idx)\}$ // Features that precede $idx$ in
        permutation $\pi$
      $M_2 \leftarrow M_1 \cup \{idx\}$
      $b_1 \leftarrow x_i[M_1]$ // Vector $b_1$ as in IME method (Štrumbelj & Kononenko (2013))
      $b_2 \leftarrow x_i[M_2]$ // Vector $b_2$ as in IME method (Štrumbelj & Kononenko (2013))
      $X \leftarrow X \cup \{(b_1, 0), (b_2, 0)\}$ // Add out of distribution instances
   **end**
**end**
$d \leftarrow dModel.fit(X)$ // Fit model *dModel* to set $X$ and save it in $d$
**return** $d$

---

dataset from UCI repository (Dua & Graff (2019)). Both datasets represent a binary classification problem. Apart from the target variable, condInd consists of 8 binary features, while Ionosphere consists of 34 numerical attributes. The condInd datasets contains 2000 instances and Ionosphere contains 351 instances.

The results are shown in Table 4. The differences between original LIME and gLIME explanations are considerable (see the top table). This is not surprising since LIME fits local linear models in its local explanations, which can strongly differ even for small perturbations of the model's input. SHAP and IME explanations are very similar (the average MAD is almost negligible). We can conclude that explanations of gSHAP and gIME are not significantly different from SHAP and IME in the non-adversary environment.

| | Perturbation | MCD-VAE | RbfDataGen | TreeEns |
|---|---|---|---|---|
| Compas_Perturbation | 0.0 | 0.676 | 1.0 | 1.0 |
| Compas_MCD-VAE | 0.997 | 0.492 | 1.0 | 1.0 |
| Compas_RbfDataGen | 0.126 | 0.006 | 0.0 | 1.0 |
| Compas_TreeEns | 0.049 | 0.144 | 1.0 | 1.0 |
| Compas2_Perturbation | 0.1 | 0.005 | 1.0 | 1.0 |
| Compas2_MCD-VAE | 0.997 | 0.322 | 1.0 | 1.0 |
| Compas2_RbfDataGen | 0.997 | 0.01 | 0.252 | 1.0 |
| Compas2_TreeEns | 0.997 | 0.036 | 0.981 | 1.0 |
| German_Perturbation | 0.0 | 0.0 | 1.0 | 1.0 |
| German_MCD-VAE | 1.0 | 0.0 | 1.0 | 1.0 |
| German_RbfDataGen | 0.33 | 0.0 | 0.0 | 1.0 |
| German_TreeEns | 0.73 | 0.0 | 0.0 | 0.92 |
| CC_Perturbation | 0.0 | 0.05 | 0.0 | 0.0 |
| CC_MCD-VAE | 0.99 | 0.26 | 0.0 | 0.0 |
| CC_RbfDataGen | 0.0 | 0.39 | 0.0 | 0.0 |
| CC_TreeEns | 0.0 | 0.4 | 0.0 | 0.0 |
| CC2_Perturbation | 0.0 | 0.065 | 0.0 | 0.0 |
| CC2_MCD-VAE | 0.985 | 0.0 | 0.0 | 0.0 |
| CC2_RbfDataGen | 0.0 | 0.075 | 0.0 | 0.0 |
| CC2_TreeEns | 0.0 | 0.07 | 0.0 | 0.0 |

| | Perturbation | MCD-VAE | RbfDataGen | TreeEns | TEnsFillIn |
|---|---|---|---|---|---|
| Compas_Perturbation | 0.426 | 0.447 | 0.841 | 0.929 | 0.798 |
| Compas_MCD-VAE | 0.409 | 0.207 | 0.667 | 0.837 | 0.782 |
| Compas_RbfDataGen | 0.57 | 0.094 | 0.104 | 0.626 | 0.51 |
| Compas_TreeEns | 0.252 | 0.209 | 0.019 | 0.202 | 0.227 |
| Compas2_Perturbation | 0.327 | 0.417 | 0.908 | 0.989 | 0.799 |
| Compas2_MCD-VAE | 0.629 | 0.528 | 0.901 | 0.96 | 0.794 |
| Compas2_RbfDataGen | 0.519 | 0.154 | 0.286 | 0.754 | 0.513 |
| Compas2_TreeEns | 0.252 | 0.259 | 0.162 | 0.497 | 0.243 |
| German_Perturbation | 0.099 | 0.188 | 0.17 | 0.28 | 0.297 |
| German_MCD-VAE | 0.27 | 0.25 | 0.81 | 0.75 | 0.61 |
| German_RbfDataGen | 0.04 | 0.078 | 0.0 | 0.0 | 0.0 |
| German_TreeEns | 0.0 | 0.049 | 0.0 | 0.0 | 0.0 |
| CC_Perturbation | 0.18 | 0.115 | 0.0 | 0.235 | 0.645 |
| CC_MCD-VAE | 0.18 | 0.155 | 0.0 | 0.045 | 0.345 |
| CC_RbfDataGen | 0.945 | 0.165 | 0.0 | 0.935 | 0.945 |
| CC_TreeEns | 0.335 | 0.27 | 0.0 | 0.0 | 0.025 |
| CC2_Perturbation | 0.21 | 0.065 | 0.0 | 0.66 | 0.665 |
| CC2_MCD-VAE | 0.115 | 0.04 | 0.0 | 0.18 | 0.37 |
| CC2_RbfDataGen | 0.955 | 0.78 | 0.0 | 0.945 | 0.96 |
| CC2_TreeEns | 0.36 | 0.325 | 0.0 | 0.04 | 0.02 |

| | Perturbation | MCD-VAE | TEnsFillIn |
|---|---|---|---|
| Compas_Perturbation | 0.77 | 0.511 | 0.772 |
| Compas_MCD-VAE | 0.948 | 0.463 | 0.877 |
| Compas_TEnsFillIn | 0.738 | 0.476 | 0.754 |
| Compas2_Perturbation | 0.94 | 0.487 | 0.77 |
| Compas2_MCD-VAE | 0.987 | 0.443 | 0.875 |
| Compas2_TEnsFillIn | 0.93 | 0.497 | 0.761 |
| German_Perturbation | 0.67 | 0.25 | 0.6 |
| German_MCD-VAE | 0.95 | 0.28 | 0.88 |
| German_TEnsFillIn | 0.78 | 0.257 | 0.644 |
| CC_Perturbation | 0.075 | 0.28 | 0.0 |
| CC_MCD-VAE | 0.96 | 0.17 | 0.86 |
| CC_TEnsFillIn | 0.11 | 0.34 | 0.0 |
| CC2_Perturbation | 0.565 | 0.15 | 0.115 |
| CC2_MCD-VAE | 0.945 | 0.05 | 0.845 |
| CC2_TEnsFillIn | 0.555 | 0.35 | 0.085 |

Table 3: The robustness results for gLIME (top table), gSHAP (middle table), and gIME (bottom table). The tables show the proportion of evaluation set instances, where the sensitive feature was recognized as the most important by the used explanation method. Columns represent the generators used for explanations. The row labels consist of the name of the dataset on which the experiment was performed and the name of the generator used for training of the adversarial model. Compas2 and CC2 denote attacks with two independent features. Perturbation represents the original sampling used in LIME, SHAP, and IME, TEnsFillIn represents a variant of the TreeEnsemble generator where new instances are generated around the given one, and TreeEns represents the generation from the whole distribution.

| | COMPAS | | condInd | | Ionosphere | |
|---|---|---|---|---|---|---|
| | Average MAE | Variance of MAE | Average MAE | Variance of MAE | Average MAE | Variance of MAE |
| Bayes_MCD-VAE | 0.1196 | 0.0029 | 0.038 | 0.0002 | 0.011 | 0.0 |
| Bayes_RBF | 0.1361 | 0.0006 | 0.0094 | 0.0 | 0.019 | 0.0002 |
| Bayes_TEns | 0.1144 | 0.0011 | 0.011 | 0.0 | 0.0223 | 0.0002 |
| SVM_MCD-VAE | 0.0206 | 0.0 | 0.0557 | 0.0005 | 0.0591 | 0.0002 |
| SVM_RBF | 0.1206 | 0.0011 | 0.0126 | 0.0 | 0.0309 | 0.0001 |
| SVM_TEns | 0.0774 | 0.0003 | 0.0147 | 0.0 | 0.0307 | 0.0001 |
| Forest_MCD-VAE | 0.1991 | 0.0079 | 0.0419 | 0.0002 | 0.0933 | 0.0009 |
| Forest_RBF | 0.1326 | 0.0023 | 0.022 | 0.0 | 0.0223 | 0.0001 |
| Forest_TEns | 0.1158 | 0.0016 | 0.0247 | 0.0 | 0.0258 | 0.0001 |
| NN_MCD-VAE | 0.0416 | 0.0001 | 0.1252 | 0.0011 | 0.1155 | 0.0007 |
| NN_RBF | 0.1623 | 0.0019 | 0.0185 | 0.0 | 0.0369 | 0.0001 |
| NN_TEns | 0.097 | 0.0006 | 0.0209 | 0.0 | 0.037 | 0.0001 |

| | COMPAS | | condInd | | Ionosphere | |
|---|---|---|---|---|---|---|
| | Average MAE | Variance of MAE | Average MAE | Variance of MAE | Average MAE | Variance of MAE |
| Bayes_MCD-VAE | 0.0307 | 0.0007 | 0.0735 | 0.0004 | 0.0099 | 0.0 |
| Bayes_RBF | 0.0533 | 0.0004 | 0.0019 | 0.0 | 0.0106 | 0.0 |
| Bayes_TEns | 0.023 | 0.0003 | 0.0069 | 0.0 | 0.0079 | 0.0 |
| Bayes_TEnsFill | 0.031 | 0.001 | 0.0483 | 0.0001 | 0.0101 | 0.0001 |
| SVM_MCD-VAE | 0.009 | 0.0001 | 0.0889 | 0.0007 | 0.0219 | 0.0002 |
| SVM_RBF | 0.0101 | 0.0002 | 0.0019 | 0.0 | 0.0125 | 0.0 |
| SVM_TEns | 0.0129 | 0.0001 | 0.0069 | 0.0 | 0.012 | 0.0 |
| SVM_TEnsFill | 0.011 | 0.0002 | 0.0483 | 0.0001 | 0.0262 | 0.0001 |
| Forest_MCD-VAE | 0.0368 | 0.0004 | 0.0895 | 0.0009 | 0.0167 | 0.0 |
| Forest_RBF | 0.0343 | 0.0003 | 0.0127 | 0.0 | 0.0132 | 0.0 |
| Forest_TEns | 0.0291 | 0.0002 | 0.0183 | 0.0001 | 0.0125 | 0.0 |
| Forest_TEnsFill | 0.0361 | 0.0006 | 0.0353 | 0.0001 | 0.0181 | 0.0 |
| NN_MCD-VAE | 0.017 | 0.0003 | 0.0577 | 0.0004 | 0.0193 | 0.0001 |
| NN_RBF | 0.0262 | 0.0003 | 0.0091 | 0.0 | 0.0111 | 0.0 |
| NN_TEns | 0.0164 | 0.0001 | 0.0147 | 0.0 | 0.0126 | 0.0 |
| NN_TEnsFill | 0.0183 | 0.0004 | 0.0364 | 0.0001 | 0.0215 | 0.0 |

| | COMPAS | | condInd | | Ionosphere | |
|---|---|---|---|---|---|---|
| | Average MAE | Variance of MAE | Average MAE | Variance of MAE | Average MAE | Variance of MAE |
| Bayes_MCD-VAE | 0.0329 | 0.0005 | 0.0052 | 0.0005 | 0.0137 | 0.0 |
| Bayes_TEnsFill | 0.0209 | 0.0003 | 0.0019 | 0.0 | 0.0093 | 0.0 |
| SVM_MCD-VAE | 0.0112 | 0.0001 | 0.0223 | 0.002 | 0.0276 | 0.0002 |
| SVM_TEnsFill | 0.0069 | 0.0001 | 0.0019 | 0.0 | 0.0235 | 0.0001 |
| Forest_MCD-VAE | 0.0333 | 0.0004 | 0.0621 | 0.0013 | 0.0151 | 0.0 |
| Forest_TEnsFill | 0.0232 | 0.0002 | 0.0349 | 0.0001 | 0.0098 | 0.0 |
| NN_MCD-VAE | 0.0202 | 0.0002 | 0.0518 | 0.0008 | 0.027 | 0.0003 |
| NN_TEnsFill | 0.0128 | 0.0001 | 0.0277 | 0.0001 | 0.0184 | 0.0001 |

Table 4: The differences in explanations between LIME and gLIME (top table), SHAP and gSHAP (middle table), and IME and gIME (bottom table). The tables show MAD and the variance of MAD on evaluation set for every combination of the explanation method and predictive model. Columns represent MAD and the variance of MAD on used datasets. The row labels consist of the used classifier and generator.

## G    ADVERSARIAL CLASSIFIERS WITH DIFFERENT PREDICTION THRESHOLDS

In a realistic scenario, the attacker might be wary of being discovered by the regulator and decide to employ deception only when it is really certain that the predicted instance is used inside the explanation method. In these circumstances, the number of times when the sensitive variable is used inside the model will also depend on how aggressively the biased classifier is used. This reflection leads to another metric for the defense's success, which measures the proportion of instances when the sensitive variable is identified when it is being used, i.e. at different levels of conservatism.

We express different levels of attacker's conservatism by changing the threshold of the decision function $d$. The output of the adversarial model $e_t$, which uses threshold $t$ is now defined with the equation:

$$e_t(x) = \begin{cases} b(x), & d_{prob}(x) \geq t \\ \psi(x), & d_{prob}(x) < t \end{cases}, \tag{8}$$

where $d_{prob}(x)$ denotes the probability that $x$ is from distribution $X_{dist}$, returned by decision model $d$. Threshold $t$ therefore represents the required certainty of the attacker that given instance is from $X_{dist}$ before it deploys the biased classifier. Larger values of $t$ will result in biased classifier being deployed fewer times but with larger certainty that it has to be used. If we use $t = 0.5$ in Equation (8), we get the adversarial model that behaves identically to the one, described with Equation (1) in Section 2.2.

In this experiment, we used only the COMPAS dataset, which we split into the training and evaluation set in ratio $90\% : 10\%$. We performed the same experiment as in section 4.2, but with different thresholds for the adversarial models. We used five different thresholds: $0.3, 0.4, 0.5, 0.6$ and $0.7$.

We first checked how different thresholds affect the deployment of the biased model. For each generator used in the adversarial model's training and the certainty threshold used by the adversarial model, we measured the proportion of instances from the evaluation set, where the biased classifier was deployed. The results are shown in Table 5. As expected, the percentage of instances where the biased classifier is deployed decreases with increasing $t$. The biased classifier is deployed less frequently when using treeEnsemble and rbfDataGen in its training, which is also expected, as these two generators give the best results inside the explanation methods. This means that for these generators, it is more difficult for an attacker to distinguish between the actual instances and generated instances. The biased classifier is deployed the most often in the case of the LIME adversarial model and least often in the SHAP adversarial model.

Next, we tested the proportion of instances where the sensitive feature (in case of COMPAS dataset this is *race*) was recognized as the most important feature by the explanation method on two different subsets of instances: instances from the evaluation set on which the biased classifier was deployed and the whole evaluation set. We measured these numbers for all combinations of thresholds and explanation methods. The results are shown in Tables 6 (gLIME), 7 (gSHAP) and 8 (gIME) for gLIME, gSHAP, and gIME, respectively. We can observe the same pattern as in section 4.2: gIME is still the most robust, and the treeEnsemble generator still gives the best results. The sensitive attribute is recognized as the most important feature more often when the decision threshold values are lower, which is expected as in these cases, the adversarial model behaves more aggressively. Whilst the percentage of the instances on which the racist behavior of the adversarial model is recognized drops with higher thresholds, it remains high enough to warn the regulator about the problematic behavior of the prediction model (especially in the case of gSHAP and gIME using treeEnsemble as data generator). From that, we can conclude that modified explanation methods remain robust enough, even with more conservative adversaries.

| | | 0.3 | 0.4 | 0.5 | 0.6 | 0.7 |
|---|---|---|---|---|---|---|
| One unrelated | Perturb. | 100.0 | 99.35 | 98.22 | 97.41 | 95.31 |
| | MCD-VAE | 98.54 | 98.38 | 98.38 | 98.22 | 97.73 |
| | RBF | 86.89 | 83.01 | 79.45 | 76.38 | 72.01 |
| | TEns | 85.28 | 74.92 | 53.88 | 27.67 | 14.4 |
| Two unrelated | Perturb. | 99.84 | 98.87 | 98.22 | 96.76 | 95.31 |
| | MCD-VAE | 99.19 | 99.19 | 99.03 | 99.03 | 98.71 |
| | RBF | 86.41 | 83.66 | 79.94 | 77.02 | 73.62 |
| | TEns | 88.03 | 77.99 | 52.27 | 28.32 | 14.72 |

| | | 0.3 | 0.4 | 0.5 | 0.6 | 0.7 |
|---|---|---|---|---|---|---|
| One unrelated | Perturb. | 88.35 | 84.79 | 79.77 | 76.05 | 70.55 |
| | MCD-VAE | 87.38 | 83.01 | 79.94 | 75.4 | 71.36 |
| | RBF | 74.27 | 64.56 | 58.25 | 49.19 | 43.37 |
| | TEns | 62.3 | 51.29 | 43.04 | 35.76 | 30.58 |
| Two unrelated | Perturb. | 89.16 | 85.28 | 82.36 | 76.7 | 70.55 |
| | MCD-VAE | 87.86 | 83.01 | 79.13 | 74.92 | 69.9 |
| | RBF | 76.05 | 66.99 | 59.55 | 53.4 | 47.09 |
| | TEns | 61.97 | 50.16 | 41.75 | 36.08 | 31.39 |

| | | 0.3 | 0.4 | 0.5 | 0.6 | 0.7 |
|---|---|---|---|---|---|---|
| One unrelated | Perturb. | 88.67 | 81.55 | 72.33 | 63.27 | 52.27 |
| | MCD-VAE | 92.72 | 89.16 | 85.44 | 79.13 | 73.3 |
| | TEnsFill | 93.2 | 87.06 | 81.72 | 72.49 | 60.84 |
| Two unrelated | Perturb. | 91.59 | 82.69 | 74.92 | 64.72 | 55.34 |
| | MCD-VAE | 94.01 | 89.81 | 85.28 | 80.58 | 74.92 |
| | TEnsFill | 93.04 | 86.25 | 78.48 | 69.74 | 60.84 |

Table 5: Proportions of instances in % of the evaluation set on which the biased classifier was deployed for adversarial LIME (top table), SHAP (middle table) and IME model (bottom table). Columns represent different threshold used for deploying the biased classifier. The rows represent the generator used in training of the adversarial attack. The labels "One unrelated" and "Two unrelated" represent the attacks with one or two unrelated features.

| | 0.3 | | 0.4 | | 0.5 | | 0.6 | | 0.7 | |
|---|---|---|---|---|---|---|---|---|---|---|
| | Biased pred. | All | Biased pred. | All | Biased pred. | All | Biased pred. | All | Biased pred. | All |
| Perturb._Perturb. | 0.0 | 0.0 | 0.0 | 0.0 | 0.0 | 0.0 | 0.0 | 0.0 | 0.0 | 0.0 |
| Perturb._MCD-VAE | 99.67 | 99.68 | 99.67 | 99.68 | 99.67 | 99.68 | 99.67 | 99.68 | 99.67 | 99.68 |
| Perturb._RBF | 99.81 | 97.9 | 99.22 | 94.01 | 25.25 | 20.71 | 0.0 | 0.0 | 0.0 | 0.0 |
| Perturb._TEns | 99.62 | 99.68 | 99.57 | 99.68 | 1.2 | 1.46 | 0.0 | 0.0 | 0.0 | 0.0 |
| MCD-VAE_Perturb. | 99.68 | 99.68 | 95.93 | 95.95 | 16.8 | 16.5 | 0.66 | 0.65 | 21.9 | 21.84 |
| MCD-VAE_MCD-VAE | 9.52 | 9.39 | 9.7 | 9.55 | 8.72 | 8.58 | 24.05 | 23.62 | 52.98 | 52.43 |
| MCD-VAE_RBF | 45.25 | 44.5 | 45.81 | 44.66 | 0.0 | 0.0 | 0.21 | 0.16 | 0.9 | 0.65 |
| MCD-VAE_TEns | 0.0 | 0.0 | 0.0 | 0.0 | 0.0 | 0.0 | 0.0 | 0.0 | 0.0 | 0.0 |
| RBF_Perturb. | 100.0 | 100.0 | 100.0 | 100.0 | 100.0 | 100.0 | 100.0 | 100.0 | 100.0 | 100.0 |
| RBF_MCD-VAE | 100.0 | 100.0 | 100.0 | 100.0 | 100.0 | 100.0 | 100.0 | 100.0 | 100.0 | 100.0 |
| RBF_RBF | 0.0 | 0.0 | 0.0 | 0.0 | 0.0 | 0.0 | 0.0 | 0.0 | 0.0 | 0.0 |
| RBF_TEns | 100.0 | 100.0 | 100.0 | 100.0 | 100.0 | 94.01 | 0.0 | 0.0 | 0.0 | 0.0 |
| TEns_Perturb. | 100.0 | 100.0 | 100.0 | 100.0 | 100.0 | 100.0 | 100.0 | 100.0 | 100.0 | 100.0 |
| TEns_MCD-VAE | 100.0 | 100.0 | 100.0 | 100.0 | 100.0 | 100.0 | 100.0 | 100.0 | 100.0 | 100.0 |
| TEns_RBF | 100.0 | 100.0 | 100.0 | 100.0 | 100.0 | 100.0 | 100.0 | 100.0 | 100.0 | 100.0 |
| TEns_TEns | 100.0 | 100.0 | 100.0 | 100.0 | 84.98 | 74.6 | 0.0 | 0.0 | 0.0 | 0.0 |

| | 0.3 | | 0.4 | | 0.5 | | 0.6 | | 0.7 | |
|---|---|---|---|---|---|---|---|---|---|---|
| | Biased pred. | All | Biased pred. | All | Biased pred. | All | Biased pred. | All | Biased pred. | All |
| Perturb._Perturb. | 98.87 | 98.87 | 56.46 | 56.8 | 11.04 | 11.33 | 2.17 | 2.1 | 0.85 | 1.13 |
| Perturb._MCD-VAE | 99.67 | 99.68 | 99.67 | 99.68 | 99.67 | 99.68 | 99.67 | 99.68 | 99.67 | 99.68 |
| Perturb._RBF | 100.0 | 99.68 | 100.0 | 99.68 | 100.0 | 99.68 | 100.0 | 99.68 | 100.0 | 99.68 |
| Perturb._TEns | 99.63 | 99.68 | 99.79 | 99.68 | 99.69 | 99.68 | 100.0 | 99.68 | 100.0 | 99.68 |
| MCD-VAE_Perturb. | 86.39 | 86.41 | 85.6 | 85.76 | 76.11 | 76.21 | 69.9 | 70.23 | 68.08 | 67.96 |
| MCD-VAE_MCD-VAE | 32.14 | 31.88 | 28.38 | 28.16 | 26.14 | 25.89 | 23.53 | 23.3 | 21.31 | 21.2 |
| MCD-VAE_RBF | 38.39 | 40.78 | 37.14 | 38.51 | 30.57 | 29.77 | 20.8 | 19.26 | 15.38 | 15.21 |
| MCD-VAE_TEns | 75.74 | 74.92 | 35.89 | 36.08 | 13.0 | 8.74 | 16.0 | 7.93 | 10.99 | 8.9 |
| RBF_Perturb. | 100.0 | 100.0 | 100.0 | 100.0 | 100.0 | 100.0 | 100.0 | 100.0 | 100.0 | 100.0 |
| RBF_MCD-VAE | 100.0 | 100.0 | 100.0 | 100.0 | 100.0 | 100.0 | 100.0 | 100.0 | 100.0 | 100.0 |
| RBF_RBF | 39.14 | 41.75 | 36.75 | 35.44 | 3.24 | 2.75 | 3.36 | 2.75 | 0.0 | 0.0 |
| RBF_TEns | 100.0 | 100.0 | 100.0 | 100.0 | 100.0 | 100.0 | 99.43 | 95.15 | 58.24 | 50.65 |
| TEns_Perturb. | 100.0 | 100.0 | 100.0 | 100.0 | 100.0 | 100.0 | 100.0 | 100.0 | 100.0 | 100.0 |
| TEns_MCD-VAE | 100.0 | 100.0 | 100.0 | 100.0 | 100.0 | 100.0 | 100.0 | 100.0 | 100.0 | 100.0 |
| TEns_RBF | 100.0 | 100.0 | 100.0 | 100.0 | 100.0 | 100.0 | 100.0 | 100.0 | 100.0 | 100.0 |
| TEns_TEns | 100.0 | 100.0 | 100.0 | 100.0 | 100.0 | 100.0 | 100.0 | 100.0 | 4.4 | 3.88 |

Table 6: Percentages of instances where the sensitive feature (*race*) was recognized as the most important feature with gLIME for adversarial attacks with one (top table) or two unrelated features (bottom table). The columns labeled *Biased pred.* represent the results on instances on which the biased classifier was deployed, while the columns labeled *All* represent the results on the whole evaluation sets. The numbers above represent the used threshold. The row labels are in the form *<explainer>_<adversarial>* where *<explainer>* denotes the generator used in the explanation method and *<adversarial>* denotes the generator used in the training of the adversarial model.

| | 0.3 | | 0.4 | | 0.5 | | 0.6 | | 0.7 | |
|---|---|---|---|---|---|---|---|---|---|---|
| | Biased pred. | All | Biased pred. | All | Biased pred. | All | Biased pred. | All | Biased pred. | All |
| Perturb._Perturb. | 41.03 | 39.0 | 31.11 | 27.99 | 24.95 | 22.01 | 21.91 | 19.09 | 16.74 | 13.11 |
| Perturb._MCD-VAE | 99.63 | 94.34 | 99.03 | 90.61 | 96.96 | 86.08 | 94.21 | 80.1 | 88.66 | 70.55 |
| Perturb._RBF | 99.35 | 90.78 | 97.99 | 82.52 | 81.39 | 59.71 | 61.18 | 39.48 | 36.94 | 20.71 |
| Perturb._TEns | 93.77 | 80.58 | 62.15 | 50.32 | 32.33 | 22.33 | 19.0 | 11.0 | 13.76 | 5.99 |
| MCD-VAE_Perturb. | 72.16 | 65.21 | 57.44 | 51.78 | 32.66 | 26.38 | 61.7 | 50.32 | 51.38 | 39.0 |
| MCD-VAE_MCD-VAE | 32.78 | 30.74 | 47.95 | 41.91 | 40.89 | 34.79 | 29.83 | 27.02 | 24.26 | 18.93 |
| MCD-VAE_RBF | 45.97 | 42.72 | 33.83 | 30.58 | 24.72 | 20.23 | 16.45 | 11.97 | 11.94 | 7.44 |
| MCD-VAE_TEns | 47.01 | 41.26 | 31.55 | 27.02 | 15.79 | 12.78 | 9.05 | 7.93 | 10.58 | 8.25 |
| RBF_Perturb. | 99.08 | 95.95 | 97.52 | 92.07 | 95.94 | 86.25 | 88.72 | 73.79 | 71.56 | 53.56 |
| RBF_MCD-VAE | 99.63 | 95.15 | 98.44 | 88.67 | 93.72 | 79.61 | 89.7 | 71.04 | 74.15 | 55.02 |
| RBF_RBF | 46.41 | 37.22 | 41.1 | 28.16 | 18.61 | 12.46 | 9.54 | 6.15 | 3.73 | 1.94 |
| RBF_TEns | 66.49 | 48.71 | 30.91 | 20.23 | 10.9 | 6.63 | 2.26 | 1.78 | 0.0 | 0.0 |
| TEns_Perturb. | 100.0 | 99.03 | 100.0 | 96.28 | 99.39 | 92.07 | 99.57 | 87.38 | 98.62 | 79.13 |
| TEns_MCD-VAE | 100.0 | 96.93 | 100.0 | 95.15 | 100.0 | 91.75 | 99.57 | 87.7 | 98.19 | 78.96 |
| TEns_RBF | 98.47 | 86.57 | 96.74 | 76.38 | 88.06 | 60.19 | 81.25 | 46.76 | 64.93 | 32.2 |
| TEns_TEns | 93.51 | 70.55 | 71.29 | 42.56 | 41.73 | 20.87 | 18.1 | 8.09 | 11.11 | 4.21 |
| TEnsFill_Perturb. | 98.35 | 92.39 | 96.37 | 87.86 | 95.94 | 82.69 | 95.74 | 79.77 | 94.5 | 72.49 |
| TEnsFill_MCD-VAE | 96.3 | 88.67 | 92.79 | 82.2 | 89.88 | 77.02 | 86.7 | 70.71 | 82.99 | 62.94 |
| TEnsFill_RBF | 86.06 | 70.87 | 74.94 | 55.83 | 65.28 | 44.34 | 57.24 | 34.79 | 45.15 | 24.92 |
| TEnsFill_TEns | 65.19 | 49.19 | 43.53 | 28.96 | 29.32 | 17.15 | 23.08 | 11.49 | 17.46 | 8.25 |

| | 0.3 | | 0.4 | | 0.5 | | 0.6 | | 0.7 | |
|---|---|---|---|---|---|---|---|---|---|---|
| | Biased pred. | All | Biased pred. | All | Biased pred. | All | Biased pred. | All | Biased pred. | All |
| Perturb._Perturb. | 66.06 | 60.03 | 57.87 | 50.0 | 41.45 | 34.79 | 24.68 | 19.26 | 13.99 | 10.19 |
| Perturb._MCD-VAE | 99.82 | 95.63 | 99.81 | 94.17 | 99.18 | 90.61 | 98.49 | 85.76 | 95.83 | 78.64 |
| Perturb._RBF | 98.94 | 92.72 | 97.1 | 83.01 | 88.04 | 66.67 | 64.55 | 41.59 | 48.8 | 27.51 |
| Perturb._TEns | 97.39 | 84.79 | 70.0 | 48.38 | 36.43 | 20.06 | 16.14 | 8.25 | 9.28 | 4.21 |
| MCD-VAE_Perturb. | 63.16 | 59.22 | 50.28 | 44.01 | 42.04 | 35.44 | 65.4 | 53.56 | 43.81 | 34.3 |
| MCD-VAE_MCD-VAE | 53.41 | 48.22 | 35.28 | 30.58 | 37.63 | 30.26 | 43.63 | 35.92 | 25.69 | 20.23 |
| MCD-VAE_RBF | 37.87 | 33.66 | 24.15 | 20.71 | 14.95 | 10.68 | 34.85 | 25.24 | 12.71 | 7.61 |
| MCD-VAE_TEns | 37.34 | 29.29 | 50.97 | 38.19 | 34.11 | 23.62 | 22.87 | 15.86 | 15.46 | 8.9 |
| RBF_Perturb. | 99.64 | 98.06 | 98.48 | 94.82 | 97.45 | 90.78 | 96.84 | 86.25 | 92.89 | 77.51 |
| RBF_MCD-VAE | 99.45 | 96.93 | 96.1 | 91.59 | 92.64 | 85.44 | 87.47 | 76.54 | 78.47 | 63.43 |
| RBF_RBF | 70.21 | 59.39 | 60.14 | 45.95 | 53.26 | 35.6 | 44.55 | 26.54 | 37.11 | 19.74 |
| RBF_TEns | 88.25 | 76.21 | 70.0 | 50.97 | 41.09 | 22.17 | 29.6 | 12.62 | 18.04 | 7.12 |
| TEns_Perturb. | 100.0 | 99.68 | 100.0 | 99.03 | 100.0 | 97.9 | 99.58 | 92.39 | 99.54 | 87.38 |
| TEns_MCD-VAE | 100.0 | 99.35 | 99.81 | 98.06 | 99.8 | 95.79 | 99.57 | 92.39 | 99.54 | 85.76 |
| TEns_RBF | 100.0 | 94.5 | 97.34 | 83.98 | 94.29 | 70.39 | 91.82 | 59.55 | 88.66 | 50.65 |
| TEns_TEns | 98.17 | 88.03 | 89.35 | 62.46 | 77.13 | 43.85 | 66.37 | 31.23 | 55.67 | 22.49 |
| TEnsFill_Perturb. | 99.09 | 94.01 | 98.1 | 89.97 | 97.05 | 85.92 | 96.84 | 79.29 | 95.41 | 72.49 |
| TEnsFill_MCD-VAE | 96.32 | 88.83 | 95.71 | 84.14 | 92.84 | 78.8 | 89.85 | 72.82 | 85.42 | 63.11 |
| TEnsFill_RBF | 85.53 | 70.87 | 73.91 | 55.5 | 66.3 | 44.66 | 59.09 | 36.25 | 51.89 | 28.16 |
| TEnsFill_TEns | 72.06 | 52.91 | 50.65 | 32.04 | 37.6 | 20.39 | 29.15 | 14.08 | 24.23 | 10.52 |

Table 7: Percentage of instances where the sensitive attribute (*race*) was recognized as the most important feature with gSHAP for adversarial attacks with one (top table) or two unrelated features (bottom table). The columns labeled *Biased pred.* represent the results on instances on which the biased classifier was deployed, while columns labeled *All* represent the results on the whole evaluation sets. The numbers above represent the used threshold. The row labels are in the form *<explainer>_<adversarial>* where *<explainer>* denotes the generator used in the explanation method and *<adversarial>* denotes the generator used in the training of the adversarial model.

| | 0.3 | | 0.4 | | 0.5 | | 0.6 | | 0.7 | |
|---|---|---|---|---|---|---|---|---|---|---|
| | Biased pred. | All | Biased pred. | All | Biased pred. | All | Biased pred. | All | Biased pred. | All |
| Perturb._Perturb. | 99.82 | 98.22 | 97.22 | 92.23 | 88.37 | 76.7 | 68.8 | 51.78 | 41.18 | 24.6 |
| Perturb._MCD-VAE | 100.0 | 98.38 | 100.0 | 96.93 | 100.0 | 95.15 | 99.59 | 89.16 | 98.45 | 80.1 |
| Perturb._TEnsFill | 97.4 | 95.15 | 93.31 | 87.54 | 88.71 | 79.61 | 79.46 | 65.05 | 69.68 | 48.22 |
| MCD-VAE_Perturb. | 50.0 | 49.68 | 57.34 | 55.83 | 59.96 | 57.61 | 50.13 | 47.73 | 33.13 | 25.24 |
| MCD-VAE_MCD-VAE | 59.16 | 57.77 | 37.93 | 34.47 | 47.35 | 41.75 | 35.17 | 30.42 | 21.19 | 16.34 |
| MCD-VAE_TEnsFill | 74.31 | 72.49 | 67.1 | 63.92 | 63.96 | 59.55 | 41.52 | 37.22 | 33.51 | 29.94 |
| TEnsFill_Perturb. | 99.45 | 95.31 | 97.62 | 89.16 | 91.72 | 76.86 | 78.52 | 58.58 | 54.49 | 33.17 |
| TEnsFill_MCD-VAE | 99.3 | 94.98 | 99.09 | 92.88 | 98.48 | 88.51 | 95.09 | 79.45 | 94.48 | 73.79 |
| TEnsFill_TEnsFill | 98.96 | 95.31 | 97.03 | 90.45 | 93.07 | 82.52 | 83.48 | 67.64 | 76.06 | 53.4 |

| | 0.3 | | 0.4 | | 0.5 | | 0.6 | | 0.7 | |
|---|---|---|---|---|---|---|---|---|---|---|
| | Biased pred. | All | Biased pred. | All | Biased pred. | All | Biased pred. | All | Biased pred. | All |
| Perturb._Perturb. | 97.0 | 95.47 | 98.24 | 95.15 | 99.14 | 94.82 | 99.25 | 94.82 | 99.42 | 94.17 |
| Perturb._MCD-VAE | 98.8 | 97.73 | 99.1 | 97.57 | 99.81 | 97.57 | 99.8 | 97.57 | 99.78 | 97.73 |
| Perturb._TEnsFill | 93.39 | 91.75 | 94.56 | 91.26 | 96.7 | 91.59 | 97.91 | 91.91 | 98.14 | 92.07 |
| MCD-VAE_Perturb. | 56.18 | 54.69 | 56.16 | 52.75 | 53.35 | 48.38 | 51.25 | 49.51 | 48.83 | 48.38 |
| MCD-VAE_MCD-VAE | 31.84 | 29.94 | 52.07 | 48.06 | 69.64 | 62.94 | 54.82 | 49.19 | 41.47 | 38.03 |
| MCD-VAE_TEnsFill | 63.65 | 62.46 | 66.98 | 62.94 | 68.66 | 65.86 | 64.5 | 59.71 | 58.51 | 57.28 |
| TEnsFill_Perturb. | 79.68 | 75.89 | 85.71 | 77.18 | 90.28 | 76.7 | 90.25 | 76.38 | 92.69 | 77.83 |
| TEnsFill_MCD-VAE | 92.6 | 88.67 | 95.32 | 88.51 | 97.91 | 88.83 | 98.39 | 88.03 | 99.14 | 88.35 |
| TEnsFill_TEnsFill | 83.65 | 80.91 | 87.43 | 80.58 | 91.75 | 80.26 | 92.81 | 80.58 | 94.68 | 80.42 |

Table 8: Percentages of instances where the sensitive attribute (*race*) was recognized as the most important feature with gIME for adversarial attacks with one (top table) or two unrelated features (bottom table). The columns labeled *Biased pred.* represent the results on instances on which the biased classifier was deployed, while the columns labeled *All* represent the results on the whole evaluation sets. The numbers above represent the used threshold. The row labels are in the form *<explainer>_<adversarial>* where *<explainer>* denotes the generator used in the explanation method and *<adversarial>* denotes the generator used in the training of the adversarial model.

