# OpenReview forum: "Better sampling in explanation methods can prevent dieselgate-like deception"
_ICLR.cc/2021/Conference — Reject_

### Official Review · AnonReviewer3 · 2020-10-21
**Nice use case for data generators, requires more analysis to contribute to explanation literature**

**Rating:** 4
**Confidence:** 4

**Review:**

Summary: This paper proposes a defense against the adversarial attacks on explanation methods described in Slack et al. (2020). In particular, by using sampling methods that more closely resemble the original data distribution, the authors make it difficult for the Out-of-Distribution detector to successfully discriminate between instances used for predicting and instances used for explaining.

Positives: The paper represents a nice use case for this suite of data generators. The necessary background information is explained well, and the testing is thorough with respect to comparing robustness across the data generators

Negatives: Firstly,  I'm concerned that the evaluation metric you use does not fully  capture the nature of the problem. In particular, it is not clear that in testing you ensure that the biased classifier is deployed on all testing instances. The % of the time that the sensitive variable appears in the top position will also depend on how aggressively the biased classifier is used. When testing/explaining points are separable, this is less of an issue, as the biased explainer will be used on all (or almost all) testing points. However, the purpose of your method is to make testing/explaining points more difficult to distinguish. In this case we might imagine that the biased explainer is being used more conservatively - that is, if the adversarial classifier cannot distinguish between testing points and sampling points and therefore deploys the unbiased classifier on some testing points, the sensitive attribute may correctly go undetected. The true metric of interest should relate to the % time the sensitive variable is identified when it is being used, at different levels of conservatism.

The authors of LIME, SHAP, and IME make careful design choices for reasons of 1) breaking data correlations, 2) ensuring the satisfaction of certain axioms, and 3) run-time. It would be important to relate this new methodology to its predecessors along these lines as well. In particular, sampling only from the manifold of the original training data can be expected to maintain the correlation structure of the original data. This is useful for fooling the adversarial classifier but you may sacrifice an ability to differentiate between the model's usage of correlated features.

I'd like to see some sensitivity analysis to the number of samples. You point out that IME is "already quite robust" but certainly this seems counterintuitive at small sample sizes. At which sample size does this become true?

Related work: Saito et al. address this problem concurrently https://arxiv.org/pdf/2006.12302.pdf

All in all, I find the work to be a useful step forward, but believe that it would benefit from more thorough analysis before publication.

---

> ### Author Response · Authors · 2020-11-24
> **Thank you for your suggestions, paper updated**
>
> Thank you for careful reading and suggestions which helped us to improve the work. We have added some additional explanations to the text and appendices of the paper.
>
> We added a new set of experiments, described in Section 4.4 and Appendix G, which addresses different levels of attackers’ conservatism. The results show that even with different thresholds, gIME is still the most robust from the three explanation methods and  treeEnsemble still gives the best results as the data generator. While the percentage of the instances, on which the biased behavior of the adversarial model is recognized, drops with higher values of the decision thresholds, it still remains high enough to warn the regulator about the problematic behavior of the prediction model (especially in the case of gSHAP and gIME using treeEnsemble as data generator).
>
> We checked how changing the certainty threshold of the decision model affects the behaviour of the adversarial model and the robustness of explanation methods. We used five different thresholds on COMPAS dataset in this experiment reported in Section 4.3 and Appendix G. The results show that gSHAP and gIME (and also gLIME, if adversarial model does not use treeEnsemble in training) are still robust enough to warn the controller about potential issues with that adversarial model.
> We checked if modified generators gIME, gLIME, and gSHAP change the obtained explanation scores. We used three dataset and five classifiers in this experiment reported in Appendix F. The results show negligible difference in explanations produced by IME and SHAP, and larger differences between LIME and gLIME.
>
> While we did no sensitivity analysis for the modified gIME method, this was done for the original IME method by Štrumbelj and Kononenko in 2010 JMLR paper (https://www.jmlr.org/papers/volume11/strumbelj10a/strumbelj10a.pdf). How to determine the correct number of samples to limit the error of the estimated Shapley values is described in section 3.2.1 of that paper (we added the reference to that paper to text in Appendix C). We used that method to determine the number of samples in our IME convergence rate experiment. The number of samples should be determined in the same way in both the adversary and non-adversary environment as it depends only on the variance of the samples.

---

### Official Review · AnonReviewer1 · 2020-10-28
**The paper suggests to replace the perturbations part for the existing post-hoc explanation methods like LIME and SHAP with on-data manifold sampling methods.**

**Rating:** 4
**Confidence:** 5

**Review:**

ICLR 2021 Review - Better Sampling in Explanation Dieselgate

Summary: The paper suggests to replace the perturbations part for the existing post-hoc explanation methods like LIME and SHAP with on-data manifold sampling methods.

SHAP and LIME use perturbations or randomly generated points to explain the decision of the black-box models. These points are out-of-distribution data, that leads to a new avenue for adversarial behavior discussed in Slack et al. (2020). The authors use existing data generators to produce better perturbations. They further empirically evaluate the robustness of explanations generated after proposed changes on real-life datasets.
Comments:
1. It is not clear what exactly the contribution of the paper. The problem is identified by existing papers [slack et al (2020), https://arxiv.org/pdf/2007.09969.pdf], etc, and mentioned that such attacks fail trivially if perturbations are from data distributions. The data generators are used from the existing literature. A recent paper [https://arxiv.org/pdf/2006.01272.pdf] proposes more efficient and theoretically sound on-data manifold SHAP computations.
2. The definition of robustness is not formally stated in the paper. The usual robustness in explanations [https://arxiv.org/pdf/1806.08049.pdf] bounded/negligible change in the explanation if the point of interest it changed slightly. It is not clear how random perturbations around the point of interest affect robustness.
3. The evaluations in the paper are weak, it is trivial that if perturbations are from data distributions the attack proposed in Slack et. al (2020) will fail (it is discussed in Slack et. al (2020) as well). Moreover, the paper does not evaluate the effects of used sampling methods in explanation. The data generating model itself a black-box model and involves more uncertainties in explanations. Minor: why one can’t use the training dataset itself to generate model explanations rather than using black-box data generators?


**After Rebuttal**

I would like to thank the authors for their rebuttal. I agree that it is not fair to assess the merits of the current work based on papers that were not available at the time of submission (or that, strictly speaking, have not been published at the time of submission). Indeed, to an extent, pointing out the ArXiv paper encourages authors to simply submit their works there to get a "publication" stamp, which on a community level is undesirable (papers on ArXiv aren't reviewed and citing them as scientific sources is problematic to say the least). I suppose the only point is that there exist works that do similar things in a more compelling fashion.

It's encouraging to see that the authors checked for robustness of their method, and I appreciate the efforts.

With these two issues resolved to a certain extent, I am willing to increase my score.

---

> ### Author Response · Authors · 2020-11-24
> **Thank you for your feedback, some additional explanations added to the paper**
>
> Thank you for careful reading and suggested references which we help us to further improve the work. We have added some additional explanations to the text and appendices of the paper.
>
> The mentioned ArXiv paper (Frye et al, 2020, Shapley explainability on the data manifold) was not available at the time of ICLR submission and we could not know about it. Further, this paper suggests a different explanation method which may or may not be more robust than the existing explanation methods (robustness was not analysed, nor adversarial attacks were prevented).
> Our paper uses the term robustness in a sense of prevention of adversarial attacks and not in a sense defined in the mentioned paper (Alvarez-Melis and Jaakkola, 2018, On the Robustness of Interpretability Methods). We now explain our use of robustness in the introduction.
>
> Concerning robustness of modified generators, we checked if modified generators gIME, gLIME, and gSHAP change the obtained explanation scores. We used three dataset and five classifiers in this experiment reported in Section 4.3 and Appendix F. The results show negligible difference in explanations produced by IME and SHAP, and larger differences between LIME and gLIME.
> The analysed perturbation-based explanation methods are standard, broadly used tools in machine learning with many uses. Improvements we propose, make them more robust. Using the training set directly to generate explanations as proposed by (Frye et al, 2020) is still open to thorough investigation and the test of time.

---

### Official Review · AnonReviewer2 · 2020-10-30

**Rating:** 4
**Confidence:** 3

**Review:**

This paper focuses on the adversarial scenario presented in Slack 2020 "Fooling LIME and SHAP: Adversarial Attacks on Post hoc Explanation Methods": an adversarial entity can design a model with obvious biases that will look innocuous to regulators when analyzed by post-hoc explanation methods such as LIME, SHAP, etc. This is achieved by leveraging the idea that the perturbations used by LIME, SHAP and other methods follow a different distribution than the original data, and therefore the adversary can learn how to distinguish the perturbed samples from the real ones and then run an unbiased version of the model when it is being probed.

This paper looks at this scenario from the eyes of the regulator that has to probe the model to decide whether it is biased. The main proposal of the paper is to alter the way in which LIME, SHAP, etc generate the perturbations needed to compute the explanation. In particular, this paper proposes to use perturbations that closely follow the data distribution, making it harder for the adversary to distinguish between genuine samples (that should go through the biased model) and perturbed samples (that should go through the unbiased model, as they imply that the model is being probed.)

Although I liked the idea exposed in the paper and enjoyed reading the background and related work, the experimental section and the conclusions interpreted from results seem a bit preliminary.

Experimentation is not very thorough, covering only robustness of the proposed sampling when pairing different generators and discriminators. The only quantitative results are provided through Figure 2 and are color coded, making them hard to compare. A table would have likely been a better way of presenting these results. More details on how the discriminator d in (eq(1)) is designed and trained would also have been of interest, particularly since different discriminators could have been evaluated.

Additional comments and questions:
- Figure 2, the use of green and red colors is inconsistent between what is described in text and figure. Text says "The green colour means that the explanation method was deceived in less than 30% of cases, and the red means that it was deceived in more than 70% of cases" but figure legend has 0 to 0.5 being red and 0.5 to 1 being green.
- One underlying assumption is that changing the perturbation used by the explanation method will not hinder the validity of the explanations. Yet, of course, explanation methods are sensitive to how the perturbations are created (trivially, one could use Gaussian noise with a very large variance to create perturbations that are not useful to generate good explanations). The paper focuses on the impact on the robustness to attacks, but more discussion and empirical results about the impact on explainability of the original method would be required.
- From section 4.2, "We consider deception successful if the sensitive feature was not recognized as the most important by the explanation method". Does that mean that deception can't be successful for samples where the sensitive feature is the most important feature on the unbiased model? Or are these features removed in the unbiased model? It is not completely clear from the explanation.
- How is the discriminator d (eq(1)) defined/trained? I could not find this information in the paper.
- Learning of the perturbation requires access to copious amounts of data from the real distribution, which may not actually be accessible to the regulator, rendering some/all of the defenses ineffective.
- "Dieselgate" is not common term?

Overall, despite the interesting idea, the paper looks to be in a preliminary state.

---

> ### Author Response · Authors · 2020-11-24
> **Thank you for your feedback, paper updated**
>
> Thank you for careful reading and informative comments which we will use to further improve our work. We have added some additional explanations to the text and appendices of the paper.
>
> Concerning the robustness results, we included Table 3 to Appendix E which includes the same information as the heatmap in Figure 2. The heatmap serves better to get a quick overview, while the tabular form gives better and more detailed information. Due to the limited space we initially included only the heatmap.
> We added a detailed description of the way how discriminators are trained to Appendix D which now includes pseudocode of three algorithms.
>
> We checked if modified generators gIME, gLIME, and gSHAP change the obtained explanation scores. We used three dataset and five classifiers in this experiment reported in Section 4.3 and Appendix F. The results show negligible difference in explanations produced by IME and SHAP, and larger differences between LIME and gLIME.
>
> We improved Section 4.2, where we now explain better how the deception is simulated. In all cases, the biased model b (see Section 2.2) was a simple function depending only on the value of the sensitive feature. The unbiased model Psi depended only on the values of unrelated random features. For unbiased models the sensitive feature was therefore never the most important one. If explanation method did not recognizing the sensitive feature, we consider the deception successful.
>
> Being aware that successful defence to adversarial attacks to explanations requires access to at least part of the training set is an important results of this paper which we stated in the conclusions: “Inspecting authorities shall be aware of the need for good data generators and make the access to training data of sensitive prediction models a legal requirement. Luckily, even a few non-deceived instances would be enough to raise an alarm about the unethical model.”

---

### Official Review · AnonReviewer4 · 2020-10-30
**Good comparison of data generators for adversarially robust explanations**

**Rating:** 7
**Confidence:** 4

**Review:**

Summary
-------------
Following the work of Slack et al (2020), which presents adversarial attacks on explanations, this work proposes a solution, that is to use improved perturbation data generators that produce instances more similar to samples in the training set.
This work also shows that the IME method is more resilient to adversarial attacks in comparison to LIME & SHAP, while both LIME and SHAP would benefit from the proposed data generators.

+ves:
-------
- Overall, the solution to use improved data generators that closely match the training data distribution is a good one including the comparison between the different data generators. The result on the robustness of IME method is good.
- Authors have submitted modified version of the code i.e. gLIME & gSHAP which use the proposed improved data generators. Both the code and the result on IME is expected to benefit the AI explainability community and practitioners that more or less rely on either LIME or SHAP today.

Possible improvements
--------------------------------
- It would be good to comment on how the data generators work on images.
- Using training data distribution may perhaps improve the overall quality of explanations as well i.e. beyond making them robust to adversarial attacks, it might be good to discuss any such benefits in the paper by considering explainability metrics such as monotonicity, faithfulness, etc.
- One thing which is unclear is author's recommendation on which data generators ( among the 3 evaluated ) to eventually use - what are their pros/cons. Does this depend on the type / distribution of data or explainability method or both.

Conclusion:
---------------
Overall, this is a nice piece of work which leverages existing data generators to show that adversarial robustness of LIME & SHAP based explanations can be improved.

---

> ### Author Response · Authors · 2020-11-24
> **Thank you for your positive attitude**
>
> Thank you for careful reading and positive attitude to our work. We have added some additional explanations to the text and appendices of the paper.
>
> The used generators were not systematically tested for images, though there are indications that they perform reasonably well. In particular, Miok et al (2019) demonstrate  their MCD VAE generator on MNIST dataset. However, there is little work using IME, LIME, or SHAP in image classification. It seems that images require specialized explanation approaches and these are mostly merged with recent neural image classifiers. We see this as an opportunity for further work.
> We checked if modified generators gIME, gLIME, and gSHAP change the obtained explanation scores. We used three dataset and five classifiers in this experiment reported in Section 4.3 and Appendix F. The results show negligible difference in explanations produced by IME and SHAP, and larger differences between LIME and gLIME.
>
> In our experiments, the instance sensitive variant of the TreeEnsemble generator, named TensFill,  worked best but we think this aspect has to be further tested using more datasets with different characteristics. E.g., for images, we presume that the MCD VAE generator or some GAN-based generator could be more successful.

---

### Decision · Program_Chairs · 2021-01-07
**Final Decision**

**Decision:**

Reject

**Comment:**

The overall impression on the paper is rather positive, however, even after rebuttal, it still seem that the paper requires further work and definitely a second review round before being ready for publication. Thus, I encourage the authors to continue with the work started during the rebuttal to address the reviewers' comment, which although moved in the right direction would still benefit from further work.  Especially, I believe the experiments could be significantly improved (by for example bringing some results to the main paper). Moreover, a more thorough comparison theoretically and empirically with previous work would increase the impact of the paper.